

# Improved BCI calibration in multimodal emotion recognition using heterogeneous adversarial transfer learning

Mehmet Ali Sarikaya and Gökhan Ince

Department of Computer Engineering, Istanbul Technical University, Istanbul, Turkey

## ABSTRACT

The use of brain-computer interface (BCI) technology to identify emotional states has gained significant interest, especially with the rise of virtual reality (VR) applications. However, the extensive calibration required for precise emotion recognition models presents a significant challenge, particularly for sensitive groups such as children, elderly, and patients. This study presents a novel approach that utilizes heterogeneous adversarial transfer learning (HATL) to synthesize electroencephalography (EEG) data from various other signal modalities, reducing the need for lengthy calibration phases. We benchmark the efficacy of three generative adversarial network (GAN) architectures, such as conditional GAN (CGAN), conditional Wasserstein GAN (CWGAN), and CWGAN with gradient penalty (CWGAN-GP) within this framework. The proposed framework is rigorously tested on two conventional open sourced datasets, SEED-V and DEAP. Additionally, the framework was applied to an immersive three-dimensional (3D) dataset named GraffitiVR, which we collected to capture the emotional and behavioral reactions of individuals experiencing urban graffiti in a VR environment. This expanded application provides insights into emotion recognition frameworks in VR settings, providing a wider range of contexts for assessing our methodology. When the accuracy of emotion recognition classifiers trained with CWGAN-GP-generated EEG data combined with non-EEG sensory data was compared against those trained using a combination of real EEG and non-EEG sensory data, the accuracy ratios were 93% on the SEED-V dataset, 99% on the DEAP dataset, and 97% on the GraffitiVR dataset. Moreover, in the GraffitiVR dataset, using CWGAN-GP-generated EEG data with non-EEG sensory data for emotion recognition models resulted in up to a 30% reduction in calibration time compared to classifiers trained on real EEG data with non-EEG sensory data. These results underscore the robustness and versatility of the proposed approach, significantly enhancing emotion recognition processes across a variety of environmental settings.

# INTRODUCTION

Emotion recognition using physiological signals has been extensively studied over the past few decades due to its potential applications in various fields such as psychology,

Corresponding author
Gökhan Ince, gokhan.ince@itu.edu.tr

healthcare, and human-computer interaction. In these studies, a variety of physiological signals has been used for emotion recognition, including electroencephalography (EEG), electromyography (EMG), electrooculography (EOG), electrodermal activity (EDA), galvanic skin response (GSR), skin temperature (SKT), respiration (RESP), blood volume pulse (BVP), heart rate (HR), and eye movements (*Bota et al., 2019*; *Soleymani et al., 2015*; *Kim & André, 2008*; *Dávila-Montero et al., 2021*; *Khalfa et al., 2002*; *Lisetti et al., 2003*; *Rainville et al., 2006*; *Nasoz et al., 2004*; *Healey & Picard, 2005*; *Zhai & Barreto, 2008*; *Hoppe et al., 2018*; *Calvo & D'Mello, 2010*).

The use of virtual reality (VR) headsets has also increased in recent years, leading to a rise in emotion-based applications (*Marín-Morales et al., 2020*). These applications include gaming, education, meditation/therapy, and physical activity. It is important to determine the emotional state of users and measure their interaction with the current virtual environment in order to enrich applications by providing personalized content. However, Emotions cannot be detected by conventional facial recognition systems using VR headsets. Instead, brain-computer interfaces (BCI) with physiological signals such as EEG, EMG, EOG, EDA, GSR, SKT, RESP, BVP, HR, and eye movements can be utilized to understand the emotions of users. Numerous studies have been conducted to detect emotions using these physiological signals (*Bota et al., 2019*). Developing successful models for emotion recognition requires a large number of subjects and trials. However, this process is tedious and costly. Given the significant time and research effort involved, it is crucial to leverage valuable insights from previously trained models for new initiatives, and to repurpose data acquired from past studies for upcoming research in VR environments.

Transfer learning methods enable the utilization of previous models and datasets. Initially, transfer learning applications have been used in image recognition and proved successful in improving recognition accuracy (*Hosna et al., 2022*). However, physiological signals are highly variable, and factors such as the individual's current mood and stress level can impact the reusability and generalization of models. Therefore, a new transfer learning approach is required to adjust models with limited data and to preserve their learning capacity despite individual differences.

Current BCIs require long calibration times for EEG-based emotion recognition algorithms (*Zhao, Yan & Lu, 2021*), and they rely on the collected EEG data from numerous training sessions for the subsequent machine learning algorithms inherently used in their design. Additionally, it is important to note that psychological patterns tend to be transient, causing classifiers to potentially underperform when applied to the same subject at a different time (*Saha & Baumert, 2020*). To build a reliable recognition model, users need to recalibrate the classifier frequently. Thus, researching adaptive emotion recognition algorithms that minimize calibration time becomes important.

In the pursuit of improved machine learning models, particularly in the domain of emotion recognition, this article introduces a novel approach leveraging the power of generative adversarial networks (GANs) (*Goodfellow et al., 2014*). GANs, which involve a duel between two neural networks to enhance prediction accuracy, have been pivotal in reducing distributional discrepancies for transfer learning applications. This technique is

particularly advantageous in scenarios necessitating swift calibration times for target domains, marking a significant stride in the realm of adversarial-based transfer learning.

The calibration time for non-EEG sensory data, such as EMG, EOG, EDA, GSR, SKT, RESP, BVP, HR, and eye movements, is notably less than for EEG sensors (*Zheng, Shi & Lu, 2020*). This article explores the feasibility of synthesizing necessary EEG data for calibration utilizing non-EEG sensory data to minimize calibration times for classifiers, especially when working with the same subject at a later time.

This research introduces several significant advancements in emotion recognition, primarily through the concept of heterogeneous adversarial transfer learning (HATL). HATL is presented as an innovative approach for generating EEG data from various modalities, specifically designed to meet the needs of EEG-based emotion recognition systems. This methodology enables effective replication of real EEG data across multiple frequency bands, reducing the need for extensive calibration sessions—a crucial benefit for applications in immersive environments such as VR. The approach includes a comprehensive comparison across three distinct GAN architectures, allowing for a detailed assessment of the synthetic data's generalizability and accuracy.

Furthermore, the study establishes a robust benchmarking framework to evaluate emotion recognition performance in both 2D and 3D environments, demonstrating the HATL architecture's efficacy in improving calibration efficiency, accuracy, and overall system performance. The generated EEG data closely aligns with real EEG in both temporal and distributional properties, leading to substantial improvements in calibration duration, a critical factor for real-world applications.

By addressing these aspects, this work sets a new standard for efficient and accurate emotion recognition within virtual and multimodal environments, showcasing the potential of transfer learning to enhance calibration processes and model reliability in both traditional 2D settings and immersive 3D contexts.

The remainder of this article is structured as follows. In the "Related Works" section, we review studies in the field of emotion recognition, emphasizing recent advances in transfer learning. The "Heterogeneous Adversarial Transfer Learning Framework for the Emotion Recognition" section describes the proposed framework for improved calibration in multimodal emotion recognition. The "Datasets and Extracted Features" section details the datasets and features utilized in our experiments. In the "Experiments" section, we outline the experimental setup and methodologies used to test our framework. The "Results" section presents the results obtained from these experiments. The "Discussion" section explores the implications of our findings and compares them with existing studies. Finally, the "Conclusion" section summarizes our contributions and suggests directions for future research.

## RELATED WORKS

Progress in sensor technology, signal processing, and Artificial Intelligence (AI), particularly in BCIs, has led to advances in emotion recognition research. Several studies focus on transfer learning to improve BCIs and emotion recognition models, but there is less research on calibration improvements. This section reviews recent studies that

examine the impact of transfer learning on BCIs and emotion recognition. Additionally, we explore the current efforts and methods aimed at enhancing calibration times and overall performance in emotion recognition.

## Background on transfer learning

In EEG-based emotion recognition, most conventional learning methods aim to improve the recognition accuracy by reducing the modelling complexity or increasing the training data using augmentation techniques. In traditional machine learning, training and test data align with the same feature space and share the same probability distribution (*Sarıkaya & İnce, 2017*; *Yasemin, Sarıkaya & Ince, 2019*). However, this hypothesis is often invalid due to the variability of human psychological signals (*Azab et al., 2018*) and the fact that EEG signals are not static, causing consistency problems (*Abbass et al., 2014*).

Transfer learning not only involves transferring models and parameters but also transfers useful information from the source domain to the target domain to reduce the domain gap in classification. It helps integrate different capable models and reduces dependence on specific data distributions and task types for EEG-based emotion recognition. Advances in transfer learning can mitigate the limitations of BCIs by removing the need for initial calibration, reducing noise in the transferred information, and decreasing the necessity for large dataset sizes.

According to the existence of labelled data, transfer learning can be divided into three categories: 1) inductive transfer learning, 2) transductive transfer learning and 3) unsupervised transfer learning (*Pan & Yang, 2009*). In inductive transfer learning, the labels of the target domain are known and the target and source tasks are different. If labelled data are available, preferably a multitask learning method is used to learn the target and source tasks simultaneously (*Hossain et al., 2017*). Otherwise, a self-taught learning technique is needed to be used (*Raina et al., 2007*). In transductive transfer learning, labeled data is available in the source domain rather than in the target domain. The target and source tasks are slightly different in their domains. If the feature spaces or marginal distributions are the same, then a sample selection bias or covariance shift method can be applied. Otherwise, a domain adaptation technique is recommended (*Jiang et al., 2017*). In unsupervised transfer learning, there is no labeled data in either the source or target domain, and the target and source tasks differ. Unsupervised transfer learning uses clustering, density estimation, and dimensionality reduction to explore the correlation between the target and source domains (*Liu & Qin, 2020*).

## Transfer learning in EEG-based classification

According to the nature of the features in the source and target domains, transfer learning can be divided into two categories: 1) homogeneous transfer learning and 2) heterogeneous transfer learning (*Pan & Yang, 2009*). In homogeneous transfer learning, the dimensions of the feature space in the source and target domains are identical. In contrast, in heterogeneous transfer learning, the dimensions of the features are different. For example, *Bird et al. (2020)* conducted an EEG classification task by initializing the EEG model's weights with pre-trained parameters from an EMG classification model. This approach

was used as an alternative to initializing EEG model weights randomly, potentially leveraging the knowledge gained from the EMG model to improve the EEG classifier's performance.

*Santana, Marti & Zhang (2019)* used multi-objective genetic programming and new fitness functions that evaluate the amount of transferability to train a cross-subject classifier to predict the stimulant according to the subject's brain activity. *Zhang, Wang & Li (2017)* developed a cross-subject model using a fuzzy-based wavelet transform technique to extract the physiological features and reduce the dimensionality of the feature vector by an inter-domain scatter matrix. *Liu et al. (2019)* proposed a cross-domain framework based on transfer component analysis (TCA). They concluded that the performance of TCA was better than logistic regression and deep learning classifiers in EEG signal analysis (*Liu et al., 2019*). *Chai et al. (2017)* proposed adaptive subspace feature matching (ASFM) strategy, which learns a linear transformation function to match the source and target subspaces with both marginal and conditional distributions. *Lan et al. (2018)* proposed the maximum independence domain adaptation (MIDA) method to reduce the intra- and inter-subject variance in multiple emotion classification tasks.

*He et al. (2018)* proposed the boosting transfer learning method to learn the multi-subject task. Their training and test data were collected on different days. Their training model was tuned using a small part of the data from the test session. Their results showed that the boosting transfer learning method was significantly better than support vector machine (SVM) and AdaBoost methods. *Li et al. (2019b)* proposed a multi-source transfer learning approach based on style transfer mapping, which reduces the EEG differences between the target domain and each source domain for emotion recognition. They used few labelled data in the calibration sessions to apply source selection and style transfer. Their experiments showed that the emotion classification accuracy on three classes was improved by 12.72% compared to the accuracy obtained by the non-transfer method on the SJTU Emotion EEG Dataset (SEED) (*Zheng & Lu, 2015*). *Joadder et al. (2019)* proposed a performance-based additive feature fusion algorithm to obtain the best subset of features for motor imagery tasks. *Lin & Jung (2017)* explored a conditional transfer learning framework for emotion classification to improve subject-specific performance without increasing the labelled data by evaluating the transferability of each sample.

*Cho et al. (2015)* applied common spatial patterns (CSP) to a session-to-session transfer strategy with a regularised spatiotemporal filter, and found that CSP performed well in a zero-training task. They used variations in background noise for feature extraction to obtain a zero-calibration framework. *Kang & Choi (2014)* proposed a non-parametric Bayesian CSP model for multi-subject learning to explore the relationship between subjects by variational inference, assuming that spatial patterns across subjects share the same latent subspace. *Azab et al. (2019)* proposed an improved CSP algorithm that assigns different weights to subjects by using Kullback-Leibler (KL) divergence to measure the similarity of different subjects using subject-specific CSP. They used a regularisation parameter to ensure that classification parameters are closely aligned with those of previous users who have feature spaces similar to the target subject. *Zanini et al. (2017)* applied Riemannian manifold geometry of symmetric positive definite matrices and

minimum distance to mean classifier for cross-subject motor imagery tasks. They transformed the covariance matrices for each subject to centre them for comparison of cross-subject data. *Nguyen, Karavas & Artemiadis (2017)* explored a method using the Riemannian manifold based on covariance matrix descriptors and the relevance vector machine classifier to measure the difference between source and target regions. Their method achieved improved accuracy and robustness compared to other approaches in the field.

## Calibration improvements in EEG-based classification

In the field of BCI, calibration refers to the initial training phase that tailors a BCI system to an individual user by collecting sufficient EEG data to achieve reliable classification accuracy. This calibration phase can often be time-consuming, causing fatigue and hindering real-time usability, especially in applications where fast setup is essential. To address this challenge, recent studies have proposed diverse calibration time reduction techniques, each yielding notable efficiency gains for different BCI paradigms.

*Zheng, Shi & Lu (2020)* proposed a heterogeneous transfer learning model to improve calibration time in cross-subject affective state detection. They extracted the scan path from eye-tracking data and used this information to reduce the calibration time from the EEG data collection session.

*Hossain et al. (2018)* developed a feature selection strategy using informative transfer learning. This approach leverages filter bank common spatial patterns to reduce the training data requirement by 58%. Despite the reduction in data, it maintains benchmark performance in multiclass BCIs (*Hossain et al., 2018*). This method combines active learning with a robust feature extraction process, significantly reducing calibration time while preserving classification accuracy.

*Jin et al. (2019)* introduced a generic model set that decreased calibration time for P300-based BCI systems by 70.7%, achieving an equivalent 80% classification accuracy compared to traditional models. By matching each new user with a pre-trained model, this approach minimizes the calibration duration without compromising system efficacy, highlighting its applicability in user-specific settings.

*Khalaf & Akcakaya (2020)* presented a probabilistic transfer learning approach for BCIs, achieving a 60.43% reduction in calibration time for motor imagery tasks by using data from similar users to enhance the training dataset. This method ensures sufficient data coverage with minimal calibration sessions, proving effective in paradigms that are challenging for users with limited motor abilities.

*Xiong & Wei (2022)* reduced calibration time by applying empirical mode decomposition and data alignment for motor imagery BCIs. Their technique generates synthetic EEG data to augment initial training sets, achieving comparable classification accuracy with two-thirds fewer trials per class.

*Xu et al. (2022)* employed a balanced Wasserstein GAN with gradient penalty to counter class imbalance in rapid serial visual presentation tasks. This method reduced the original data requirement by 40% while achieving a classification area under the curve (AUC)

increase of 3.7%, demonstrating the potential of GAN-based data augmentation in calibration time reduction across BCI applications.

## Transfer learning with deep learning

A deep neural network (DNN) is a machine learning method consisting of a neural network with an input layer, an output layer, and several hidden layers. This method is often applied to EEG-based emotion recognition tasks (*Sarıkaya & İnce, 2017*). DNN is also used in several transfer learning tasks such as network adaptation, feature transfer and parameter transfer. Convolution-based transfer learning, such as convolutional neural network (CNN), reuses pre-trained parts in the source domain and adversarial-based transfer learning uses adversarial technology, such as GAN. *Zhang, Zheng & Lu (2017)* proposed a subject-independent approach to evaluate sleep quality, and applied KL divergence to deep autoencoders to calculate the difference between the source and target domains in feature extraction. Their experimental results showed that the deep transfer learning model achieved better classification accuracy compared to the baseline which is an SVM-based model. *Özdenizci et al. (2020)* proposed an adversarial inference approach to reduce inter-subject variability on a publicly available motor imagery EEG dataset. *Li et al. (2019a)* proposed a domain adaptation method to obtain the cross-subject emotion recognition models, using adversarial training to adapt the marginal distributions in the early layers of the DNN model and performing association reinforcement to adapt the conditional distributions in the final layers.

## Deep learning in multimodal emotion recognition

Recent advancements in HCI have leveraged deep learning models for effective multimodal emotion recognition. *Muhammad, Hussain & Aboalsamh (2023)* presented a bimodal emotion recognition approach combining EEG signals and facial expressions using a Deep Canonical Correlation Analysis model. This two-stage framework first extracts modality-specific features using CNN-based architectures (ResNet50 for facial data and 1D-CNN for EEG data) and then fuses these features at the classification stage, achieving accuracies of 93.86% on the MAHNOB-HCI dataset and 91.54% on DEAP. *Wang et al. (2023)* expanded on multimodal recognition by incorporating an attention mechanism within a CNN framework to enhance critical feature extraction from facial expressions and spatial features in EEG signals. Their model achieved high classification accuracy on the DEAP and MAHNOB-HCI datasets, reporting up to 97.15% for arousal and 96.63% for valence. The attention mechanism in conjunction with feature-level fusion significantly boosted classification effectiveness in distinguishing between emotional states. *Mutawa & Hassouneh (2024)* focused on a real-time multimodal system for patient emotion recognition, especially useful for individuals with communication difficulties. By integrating geometric facial features with four-band EEG power spectral information, their model achieved 99.3% accuracy through an LSTM classifier, underscoring the robustness of multimodal fusion for reliable emotion recognition in healthcare applications.

*Jaswal & Dhingra (2023)* approached multimodal emotion recognition by analyzing EEG and audio signals. Using Principal Component Analysis and the Grey Wolf

optimization algorithm for feature selection, their CNN-based model reached an accuracy of 94.44%. Their findings emphasize the potential of audio and EEG fusion in enhancing the emotional intelligence of human-computer interaction systems, especially within the affective computing domain. *Saffaryazdi et al. (2022)* proposed an innovative approach that utilizes facial micro-expressions in combination with EEG and physiological signals, addressing the limitations of facial macro-expressions in capturing genuine emotions. Their model employs landmark-based spotting to identify regions of interest in facial data, extracting features from micro-expressions and physiological signals. Through fusion and classification, their system demonstrated that facial micro-expressions, coupled with EEG and GSR signals, can accurately recognize emotional states, highlighting the robustness of this approach for human-computer interaction and healthcare applications. *Fu et al. (2023)* developed a Multimodal Feature Fusion Neural Network (MFFNN) that integrates EEG and eye movement data. Their model employs a dual-branch feature extraction module with a multi-scale feature fusion mechanism, enhanced by cross-channel soft attention. This design enables the network to capture and fuse spatial and temporal information across modalities effectively, resulting in an accuracy of 87.32% on the SEED-IV dataset for emotions such as happiness, sadness, fear, and neutrality. The results indicate that MFFNN can capitalize on the complementary nature of EEG and eye movement data, setting a promising standard for future multimodal emotion recognition systems.

*Fu et al. (2022)* focused on addressing the challenge of individual variability and noise in physiological signals with their proposed algorithm, which uses domain adaptation to align feature distributions between source and target data. Their approach, tested on the DEAP dataset, improved emotion recognition by overcoming noise-related issues and individual differences, achieving 63.6% and 64.4% accuracy for valence and arousal, respectively. The use of substructure-based domain adaptation allows for more accurate feature projections, enhancing the algorithm's reliability across different subjects. *Moin et al. (2023)* introduced a framework utilizing EEG and facial gesture data for emotion recognition. Their method extracts spectral and statistical features from EEG signals and histogram of oriented gradients and local binary pattern features from facial images. Utilizing an ensemble classification approach, they achieved high accuracy rates of 97.25% for valence and 96.1% for arousal on the DEAP dataset, demonstrating the effectiveness of multimodal approaches for cross-subject emotion recognition. *Pan et al. (2023)* proposed a deep learning-based framework, Deep-Emotion, which integrates features from facial expressions, speech, and EEG data. This approach incorporates an improved GhostNet for facial data, a lightweight fully convolutional neural network for speech signals, and a tree-like LSTM model for EEG data. Decision-level fusion was used to combine the outputs from these three modalities, achieving comprehensive performance improvements. Tested on multiple datasets (CK+, EMO-DB, and MAHNOB-HCI), the model showed robust results, making it a pioneering attempt to unify facial, speech, and EEG data for multimodal emotion recognition. *Yang et al. (2021)* focused on mobile emotion recognition using a combination of behavioral and physiological data collected from mobile and wearable devices. Their attention-based LSTM model analyzes data from

smartphone and wristband sensors, capturing facial expressions, speech, keystrokes, and various physiological signals such as blood volume and skin temperature. This model achieved an accuracy of 89.2% for binary emotion classification in real-time settings, underscoring the potential of mobile devices for in-the-wild emotion monitoring and regulation.

## Generative adversarial networks in EEG data generation

EEG data generation and augmentation has gained momentum, with various studies demonstrating the potential of GANs to improve the quality and diversity of EEG-based datasets. *Hartmann, Schirrmeister & Ball (2018)* were among the first to use GAN to generate original EEG signals, focusing on enhancing the stability and quality of the generated data. They introduced a modified Wasserstein GAN (WGAN) to ensure similar performance between the generator and the discriminator, producing single-channel EEG data that closely matched real data in both time and frequency domains. *Bouallegue & Djemal (2020)* explored a similar approach using a deep learning-based WGAN framework to generate raw EEG data. This method was validated by evaluating the performance of multiple classifiers, including multi-layer perceptron, SVM, and K-nearest neighbors, which demonstrated significant improvements in accuracy with the generated data. Their experiments, conducted with a dataset related to autism pathology, revealed that data augmentation using WGAN led to enhanced classification results.

## Generative adversarial networks in EEG-based emotion recognition

The utilization of GANs in EEG-based emotion recognition has emerged as a pivotal area of research, reflecting their increasing influence and application in this field. Table 1 presents a summary of recent studies employing GANs for this purpose, detailing the specific types of GANs used, the datasets on which they were tested, and the methods applied.

*Luo & Lu (2018)* introduced a conditional Wasserstein GAN (CWGAN) framework specifically for EEG-based emotion recognition. The study generated differential entropy features with labels to conduct data augmentation on the EEG dataset, improving the accuracy of emotion recognition models when high-quality generated data were added to the original dataset. They validated their approach using two public EEG datasets and observed a noticeable increase in performance. *Zhang et al. (2021)* proposed a multi-generator CWGAN to obtain a more diverse and comprehensive set of EEG features. This method enabled the generation of high-quality artificial data with greater feature variety, improving the performance of emotion recognition classifiers when compared to traditional single-generator GAN frameworks.

*Bao et al. (2021b)* developed a Two-level Domain Adaptation Neural Network (TDANN) that extracted deep topological features from EEG signals, achieving significant improvements in cross-day and cross-subject emotion recognition. In another contribution by *Bao et al. (2021a)* a data augmentation model using a dual discriminator GAN, named VAE-D2GAN, was introduced. This model was proficient in generating artificial EEG samples that led to increased recognition accuracies, demonstrating the

**Table 1 GANs in EEG-based emotion recognition.**

| Authors | Year | GAN type | Dataset | Methods used |
|---|---|---|---|---|
| *Luo & Lu (2018)* | 2018 | CWGAN | SEED, DEAP | Data augmentation with differential entropy |
| *Zhang et al. (2021)* | 2021 | CWGAN | SEED | Multi-generator data augmentation |
| *Bao et al. (2021b)* | 2021 | TDANN | SEED | Two-level domain adaptation neural network |
| *Bao et al. (2021a)* | 2021 | VAE-D2GAN | SEED (I–IV) | Dual discriminator GAN for data augmentation |
| *Kalashami, Pedram & Sadr (2022)* | 2022 | CWGAN | DEAP | Data augmentation and feature extraction |
| *Gu et al. (2023)* | 2023 | GAN-GCNN | DEAP, SEED | Generative graph network with GCNN |
| *Esmaeili & Kiani (2024)* | 2024 | CGAN | SEED | Emotional facial reconstruction with CGAN |
| *Qiao et al. (2024)* | 2024 | GAN | SEED | Attention-enhanced GAN for feature normalization |

potential of GANs in enhancing EEG-based emotion recognition systems (*Bao et al., 2021a*). Further advancements were made by *Kalashami, Pedram & Sadr (2022)* who applied a similar CWGAN approach for data augmentation, aiming to address the challenges posed by the high dimensionality and limited quantity of EEG data. Their method showed notable improvements in the classification accuracy for emotion recognition (*Kalashami, Pedram & Sadr, 2022*).

*Gu et al. (2023)* employed a hybrid model that used generative adversarial networks and graph convolutional neural networks (GCNN) for effective emotion recognition, providing a promising direction for future research in EEG-based emotion detection systems. *Esmaeili & Kiani (2024)* presented an innovative approach by integrating emotional EEG signals with conditional GANs to generate personalized facial expressions, thereby bridging the gap between neural activities and facial expressions in emotion recognition. Similarly, *Qiao et al. (2024)* introduced a model combining attention mechanisms with GANs to enhance the robustness and accuracy of EEG emotion recognition systems, highlighting the utility of attentional strategies in managing the complexities of EEG data.

The existing literature on EEG-based emotion recognition using GANs primarily concentrates on augmenting the size of EEG datasets and enhancing the performance of classifiers. These studies typically utilize synthetic EEG data derived from existing EEG records to train more robust models. However, our approach diverges from this trend by employing transfer learning that leverages different modalities to extract EEG data, particularly tailored for 3D VR environments. Furthermore, notable studies in the field, such as those by *Zheng, Shi & Lu (2020)* and *Bird et al. (2020)*, demonstrate an application of heterogeneous transfer learning to singular modalities—eye tracking and EMG, respectively. This approach, while innovative, does not encompass the multi-modal data integration crucial for comprehensive emotion recognition in VR settings.

To address the limitations of prior studies that focused on transfer learning within the same modalities, used only non-immersive data, or worked with a single modality, we took a broader approach. We utilized the HATL framework to generate EEG-like data from multi-modal non-EEG sensory data. This approach allows us to focus on reducing calibration duration while maintaining emotion recognition accuracy in both traditional 2D datasets and immersive 3D environments.

## HETEROGENEOUS ADVERSARIAL TRANSFER LEARNING FRAMEWORK FOR EMOTION RECOGNITION

The GAN is a dual-structured neural network system comprising a generator and a discriminator (*Goodfellow et al., 2014*). The generator is tasked with producing synthetic samples that emulate the distribution of given input data, while the discriminator's role is to discern the authenticity of the samples against actual input data. The HATL architecture leverages the GAN's capability to produce EEG-like data while enabling knowledge transfer across different modalities with distinct feature types. This approach significantly reduces the need for extensive EEG data collection for new participants, which is often time-consuming and resource-intensive. The efficiency gained is particularly valuable for applications requiring rapid calibration or those where large-scale EEG data collection is impractical.

### System architecture

As depicted in Fig. 1, the proposed GAN-based HATL approach generates synthetic EEG signals from different modalities, leveraging calibration data from training subjects. The generator, trained with data representing non-EEG sensory data, learns to replicate EEG data typically obtained during calibration.

As illustrated in Fig. 1, the proposed GAN-based HATL system architecture consists of several modules, each with a specific role. Below is a brief overview of these modules and their roles:

- **Generator:** This module is responsible for generating synthetic EEG signals from non-EEG sensory data. It is trained using calibration data from various training subjects.
- **Discriminator:** The discriminator evaluates the synthetic EEG signals to determine their authenticity. It provides feedback to improve the generator's performance.
- **Trained generator:** This module generates synthetic EEG signals using non-EEG sensory data from test subjects.
- **Classifiers for emotion recognition:** These classifiers are trained to recognize emotions based on the generated EEG signals with non-EEG sensory data.
- **Trained classifiers for emotion recognition:** These classifiers predicts the emotional state of test subjects using a combination of raw EEG data and non-EEG sensory data.

This architecture emphasizes the feature-based nature of the input data, where the generator synthesizes EEG-like signals based on key characteristics extracted from non-EEG sensory inputs. Instead of utilizing raw time-series data, the non-EEG inputs are pre-processed to extract meaningful attributes, such as statistical or frequency-domain features. This ensures that the generator focuses on capturing patterns essential for generating realistic EEG signals. The use of feature-based data as input not only optimizes the generator's capacity to model neural activity but also aligns with the representation provided by frequency-based brain activation images, which serve as a concise and interpretable visualization of neural patterns rather than raw signal comparisons.

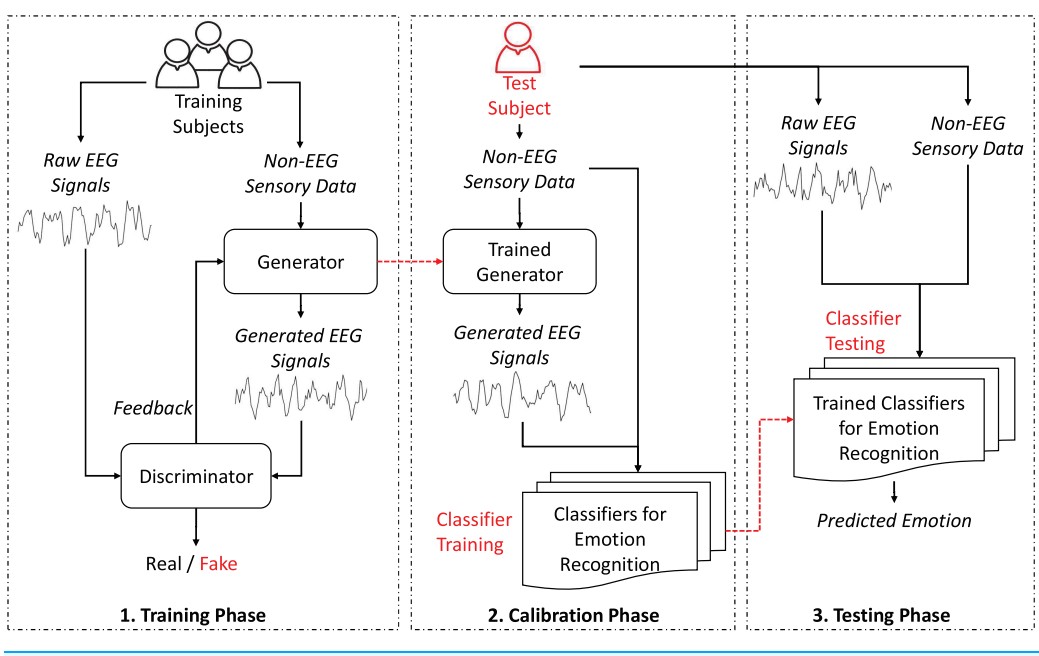

**Figure 1  System architecture.**

Further details about these modules and phases are provided in the following sections. Operational phases section discusses the proposed stages, including the training, calibration, and testing phases. Generative adversarial network models section details the interaction between the generator and the discriminator, explaining their internal functions and mechanisms. Emotion recognition models section describes the classifiers used to train and predict emotions using EEG and non-EEG sensory data.

## Operational phases
### Training phase
The process starts by collecting participants' raw EEG signals and the non-EEG sensory data. During this initial stage, the primary goal is to train the generator, ensuring it can map the non-EEG sensory data to EEG signals. The generator, leveraging the non-EEG sensory data, aims to produce EEG-like signals that closely mimic the raw EEG data. Simultaneously, the discriminator differentiates between the synthetic EEG signals produced by the generator and the actual EEG signals. The iterative feedback mechanism between the generator and discriminator enhances the similarity of the generated EEG signals to actual EEG data. The successful training of the generator affects the subsequent phases, particularly the Calibration Phase, where the model's ability to generate realistic EEG data from the non-EEG sensory data has a significant impact on the accuracy of the emotion recognition classifiers.

### Calibration phase
The calibration phase utilizes the trained generator from the training phase. During this phase, the generator creates EEG signals from the non-EEG sensory data of the test subject,

removing the need to collect new EEG data. This approach shortens calibration time since non-EEG sensory data collection requires less time compared to EEG data collection. Then, the emotion recognition classifiers are trained using the generated EEG signals and the non-EEG sensory data. This strategy prepares the emotion recognition classifiers for test subjects without extensive EEG data collection.

### Testing phase

In the testing phase, the system collects both raw EEG and non-EEG sensory data from the test subject. The emotion recognition classifiers trained during the calibration phase are then used to predict the test subject's emotional state. This phase also involves evaluating the accuracy of these classifiers. The accuracies of the emotion recognition classifiers trained on generated EEG data and non-EEG sensory data will demonstrate the effectiveness of the HATL approach in reducing calibration time while maintaining accuracy in emotion recognition.

## Generative adversarial network models

GANs consist of two neural networks: the generator and the discriminator (*Goodfellow et al., 2014*). The generator creates synthetic data, while the discriminator evaluates whether the data is real or generated. CGAN prepares the generation and discrimination processes on additional information, such as labels (*Mirza & Osindero, 2014*). Let $s$ represent the non-EEG sensory data. Our GAN models combine $s$, random noise $z$, and labels $y$ as inputs to the generator. This configuration generates EEG signals that are both realistic and aligned with specific emotional states indicated by the labels. The discriminator takes real EEG data $x$ and labels $y$ to determine whether the EEG data is real or generated.

In the proposed CGAN Eq. (1), $G$ represents the generator, $D$ represents the discriminator, $x$ represents real EEG data, $z$ is random noise, $s$ represents non-EEG sensory data, $y$ represents labels, $p$ represents distribution function, $G([z, s, y])$ is the generator output conditioned on the noise, non-EEG sensory data, and labels, and $D(x, y)$ is the discriminator's output based on real EEG data and labels.

$$\min_{G}\max_{D}(\mathbb{E}_{x,y\sim p_{\text{data}}(x,y)}[\log D(x,y)] + \mathbb{E}_{z,s,y\sim p_z(z),p_s(s),p_y(y)}[\log(1 - D(G([z, s, y]), y))]) \quad (1)$$

GANs can sometimes encounter issues with training stability and vanishing gradients (*Saxena & Cao, 2021*). The WGAN addresses these challenges by introducing the Wasserstein distance as the loss function, which improves the GAN's training dynamics and stability (*Arjovsky, Chintala & Bottou, 2017*). WGAN is a variant of GAN that minimizes the Wasserstein distance, providing a more stable optimization process (*Villani, 2009*). The Wasserstein distance, defined in Eq. (2), represents the earth-mover's distance, which measures the optimal cost of transforming one distribution into another.

$$W(P_{\text{data}}, P_g) = \inf_{\gamma \in \Pi(P_{\text{data}}, P_g)} \mathbb{E}_{(x,y)\sim\gamma}[||x - y||]. \quad (2)$$

In Eq. (2), $\Pi(P_{\text{data}}, P_g)$ represents the set of all joint distributions formed by combining $P_{\text{data}}$, the probability distribution of the real data, and $P_g$, the probability distribution of the generated data. The joint distribution is denoted by $\gamma(x, y)$, and $||x - y||$ is the distance between two samples within the joint distribution. The expected value of the distance between the samples, calculated across all possible joint distributions, gives the Wasserstein distance (*Arjovsky, Chintala & Bottou, 2017*). The WGAN model, using the Wasserstein distance, enhances training stability, reducing the likelihood of gradient disappearance. The CWGAN adds label information to condition the generation and discrimination processes. While the proposed CWGAN Eq. (3) may resemble the structure of the CGAN, it differs because the CWGAN uses the Wasserstein distance as the loss function, leading to different training dynamics.

$$\min_G \max_D (\mathbb{E}_{x,y \sim p_{\text{data}}(x,y)}[D(x,y)] - \mathbb{E}_{z,s,y \sim p_z(z),p_s(s),p_y(y)}[D(G([z,s,y]),y)]). \tag{3}$$

The CWGAN with Gradient Penalty (CWGAN-GP) extends the CWGAN by adding a gradient penalty term to ensure the Lipschitz continuity of the discriminator (*Gulrajani et al., 2017*). This addition helps stabilize the training process and reduce the chance of training-related issues like vanishing gradients or mode collapse. The proposed CWGAN-GP Eq. (4) has a similar structure to that of CWGAN, differing only in that it includes $||\nabla_{\hat{x}} D(\hat{x}, y)||_2$ as the gradient penalty term, $\lambda$ as the gradient penalty coefficient, and $\hat{x}$ as an interpolated sample between real and generated EEG data.

$$\min_G \max_D (\mathbb{E}_{x,y \sim p_{\text{data}}(x,y)}[D(x,y)] - \mathbb{E}_{z,s,y \sim p_z(z),p_s(s),p_y(y)}[D(G([z,s,y]),y)]$$
$$+ \lambda \mathbb{E}_{\hat{x},y \sim p_{\hat{x}}(\hat{x})}[(||\nabla_{\hat{x}} D(\hat{x}, y)||_2 - 1)^2]). \tag{4}$$

This gradient penalty term ensures the discriminator adheres to Lipschitz continuity, stabilizing the training process for CWGAN-GP. This setup helps improve the overall robustness of conditional adversarial networks, facilitating better performance and training stability for EEG-based applications (*Hwang et al., 2019*).

## Emotion recognition models

Our investigation into emotion recognition leverages a comprehensive analysis across three data combinations: generated EEG with the non-EEG sensory data, real EEG with the non-EEG sensory data, and exclusively the non-EEG sensory data. When using combinations involving EEG data, a feature-level fusion approach was applied, except in cases where only the non-EEG sensory data were used. To standardize input data across all models, we implemented feature normalization through MinMaxScaler, adjusting all feature values to a uniform scale from 0 to 1.

We explored the effectiveness of three prominent models in the domain of emotion recognition, each selected for their reputed performance in classification tasks:

**1) Support vector machines (SVM):** SVM is a supervised learning model renowned for its proficiency in classification and regression tasks (*Cervantes et al., 2020*). The model operates by delineating an optimal hyperplane in a high-dimensional space to distinctly

classify the data points with the widest possible margin. For our study, the SVM was configured with a radial basis function (RBF) kernel, a regularization parameter $C$ set to 1.0, and a gamma value of 'scale'.

**2) Random forests (RF):** RF employs an ensemble learning method that aggregates the outputs of numerous decision trees to boost the prediction accuracy (*Breiman, 2001*). Each tree in the forest is trained independently on a random subset of the data. The final prediction is achieved by averaging these individual outcomes, enhancing the model's generalizability and robustness. In our configuration, the RF model utilized 100 trees, with no limit on the maximum depth of the trees, and the 'gini' criterion for measuring the quality of a split.

**3) Multi-layer perceptron (MLP):** The MLP is a foundational neural network architecture consisting of layers of interconnected neurons arranged in sequence (*Gardner & Dorling, 1998*). This structure allows MLPs to address both classification and regression challenges effectively. Inputs traverse through these layers, ultimately producing the output *via* the final layer. Our MLP model was structured with one hidden layer comprising 100 neurons, employing a 'relu' activation function, and optimized using the 'adam' solver with a learning rate initially set to 0.001 and a maximum of 200 iterations for convergence.

These models were chosen for their distinct approaches to learning and their capacity to model complex relationships within the data, providing a broad perspective on the potential of each method in the context of emotion recognition. By examining the performance across different data combinations and utilizing default parameters for SVM, RF, and MLP models, our study aims to discern the most effective strategies for emotion detection and classification.

## DATASETS AND EXTRACTED FEATURES

### SEED-V

The SEED-V dataset, as established by *Liu et al. (2021)*, uses eye-tracking glasses to monitor eye movements and records EEG signals from 16 participants. This dataset includes a wide range of eye movement features, such as pupil diameter, fixation, saccades, and blinks. The detailed list of these features is provided in Table 2. EEG signal processing involves downsampling the raw data to 200 Hz, applying a bandpass filter within the 1–75 Hz range, and extracting differential entropy (DE) features across five frequency bands. The dataset comprises 310 dimensions from 62 EEG channels, 33 eye movement dimensions, and five emotional class labels including Happy, Sad, Disgust, Neutral, and Fear. For SEED-V dataset, HATL framework uses eye movement data as the non-EEG sensory data.

### DEAP

The DEAP dataset includes physiological and EEG recordings from 32 participants, who were exposed to 60-s video stimuli designed to elicit emotional responses. These recordings are categorized into several dimensions, namely arousal, valence, like/dislike, dominance, and familiarity, as documented in (*Koelstra et al., 2011*). The EEG data,

**Table 2 Summary of eye movement features in SEED-V.**

| Eye movement parameters | Extracted features |
| --- | --- |
| Pupil diameter (X and Y) | Mean, standard deviation, DE in four bands (0–0.2 Hz, 0.2–0.4 Hz, 0.4–0.6 Hz, 0.6–1.0 Hz) |
| Disperson (X and Y) | Mean, standard deviation |
| Fixation duration (ms) | Mean, standard deviation |
| Blink duration (ms) | Mean, standard deviation |
| Saccade | Mean and standard deviation of saccade duration and saccade amplitude |
| Event statistics | Blink frequency, fixation frequency, fixation duration maximum, fixation duration maximum, fixation dispersion total, fixation dispersion maximum, saccade frequency, saccade duration average, saccade amplitude average, saccade latency average. |

captured from 32 channels, span the theta (4–8 Hz), alpha (8–12 Hz), beta (12–30 Hz), and gamma (30–45 Hz) frequency bands. Additionally, the dataset encompasses a range of non-EEG sensory data which consists of horizontal EOG, vertical EOG, zygomaticus major EMG, trapezius EMG, GSR, RESP amplitude, BVP *via* plethysmograph, and SKT.

For our analysis, we selected the initial 16 subjects from the dataset, which is compliant with SEED-V. The emotional responses were classified within the valence-arousal space, segmented into four quadrants: HVHA, HVLA, LVLA, and LVHA, based on the High (H) or low (L) ratings of valence (V) and arousal (A). These ratings, which vary from 1 to 9, were divided using the median value as the threshold to categorize the responses into the aforementioned quadrants.

In terms of EEG signal processing, we extracted DE features specifically from the theta, alpha, beta, and gamma bands. This extraction was performed across all 32 EEG channels, utilizing 3-s time windows with 2-s overlaps, targeting the power spectrum of EEG signals sampled at a rate of 128 Hz.

Furthermore, for the processing of the non-EEG sensory data, we extracted eight distinctive features: mean, standard deviation, minimum, maximum, first differences, second differences, power spectrum, and average gradient. This procedure mirrors the EEG windowing approach, employing 3-s time windows with 2-s overlaps. Consequently,

the feature extraction resulted in 128 features for EEG and 64 features for the non-EEG sensory data, offering a comprehensive view of the physiological and emotional states of the participants.

## GraffitiVR

The GraffitiVR dataset is a meticulously constructed collection of data designed to analyze the relationship between human emotional responses and their behavioral manifestations in response to urban graffiti, utilizing VR to create immersive experiences for participants (*Karakas et al., 2024*). This study engaged 39 participants, focusing on emotional reactions, particularly fear and pleasure, elicited by facial expressions observed in graffiti.

For EEG data collection, this study leverages the Looxid Link package for Vive, developed as an integrated accessory device for the HTC Vive VR headset. This device is wireless, dry, and features six channels. It captures brainwave frequencies including delta (1–3 Hz), theta (3–8 Hz), alpha (8–12 Hz), beta (12–38 Hz), and gamma (38–45 Hz) from the brain's prefrontal area through channels AF3, AF4, AF7, AF8, Fp1, and Fp2 at a sampling rate of 500 Hz (*Jo & Chae, 2020*).

Integration of EEG data acquisition and monitoring with VR technology was achieved by utilizing the Looxid Link's compatibility with the HTC Vive system. The Looxid Link 2D Visualizer, an application provided with the device, allows users to observe their brain activity in real time. For the purposes of this study, we customized this application to include data logging capabilities necessary for data collection and analysis. The EEG device collects electrical brain activity and logs this activity to a document while the users are exposed to stimuli in the VR environment. Additionally, the data from the Looxid EEG sensor are saved alongside corresponding timestamps, enabling synchronization between EEG data and VR screen recordings.

To quantify the electrical activity across these bands, we computed the average band powers using Welch's periodogram (*Welch, 1967*). This involved the aggregation of areas under parabolas that were fitted to the power spectral density estimates for each frequency band, thus providing a precise measure of the power within each EEG signal frequency band.

In the GraffitiVR dataset, we selected the initial 16 subjects from the dataset, which is compliant with SEED-V and DEAP. Emotional responses were evaluated within the Valence-Arousal framework, categorized into four quadrants: HVHA, HVLA, LVLA, and LVHA, which aligns with the labels in the DEAP dataset. Ratings, ranging from 1 to 7, were segmented by the median score to systematically allocate the responses into the described quadrants.

The behavioral data within the dataset were meticulously recorded through video analysis and subsequently analyzed using the robust Lucas/Kanade optic flow algorithm (*Bruhn, Weickert & Schnörr, 2005*). This technique was instrumental in extracting detailed patterns of head movements. The analysis yielded a comprehensive set of 15 features derived from the changes in the yaw, pitch, and roll directions of head movements, including the minimum, maximum, mean, median, and standard deviation for each direction, totaling 15 features. These behavioral metrics offer a granular view of how

participants' physical responses align with their emotional experiences. For the GraffitiVR dataset, HATL framework uses head movements as the non-EEG sensory data.

In total, the EEG portion of the dataset leverages 35 features, encompassing 30 features from the five frequency bands across six channels and an additional five features including attention, relaxation, asymmetry, left brain activity, and right brain activity. Combined with the 15 features from head movement analysis, the dataset provides 50 features for exploring the dynamic interplay between emotional states and corresponding physical responses within VR environments.

## EXPERIMENTS

### Hardware and software

Development was performed on a laptop equipped with an Intel Core i7-12650H CPU, 16GB RAM, and an NVIDIA GeForce RTX 3060 GPU. The software tools used for implementation included Python 3.10, PyTorch 2.1 for model development, NumPy 1.23 for data handling, Matplotlib 3.7 for visualization, Scikit-learn 1.2.1 for evaluation metrics, mne 1.4.2 for EEG processing, and Visual Studio Code as the primary development environment. The experimental code was executed on Google Colab with a GPU runtime, utilizing a Tesla T4 GPU with 15 GB of VRAM and 12 GB of system RAM.

### HATL parameters

The experimental framework evaluates the efficacy of the proposed HATL architecture, as depicted in Fig. 1. This setup uses a standardized approach to generate synthetic EEG signals across different datasets, aiming to streamline the development of emotion recognition models by reducing the need for extensive EEG data collection.

The proposed HATL architecture synthesizes EEG-like data from the non-EEG sensory data through a Generator, which is then evaluated against real EEG data by a Discriminator. This process enhances the Generator's output for subsequent use in training an Emotion Recognition Classifier. The framework is tested across three GAN variants: CGAN, CWGAN, and CWGAN-GP, each offering different levels of stability and signal fidelity.

The key hyperparameters across all datasets include a learning rate of $2 \times 10^{-4}$, weight decay of $1 \times 10^{-6}$, and optimizer betas of $(0.5, 0.9)$ (*Luo et al., 2020*). These were selected heuristically to optimize the training process, ensuring model stability and preventing overfitting.

#### Network architectures

The HATL framework is applied to three datasets: SEED-V, DEAP, and GraffitiVR, with Generator and Discriminator configurations tailored to the complexity of the input features and the requirements of the emotion recognition tasks.

**Generator architecture:** The Generator is designed using a series of residual blocks to capture the complex patterns inherent in EEG signals derived from non-EEG data. Each residual block consists of a linear transformation followed by batch normalization and a ReLU activation function. The output of each residual block is concatenated with its input,

enhancing feature propagation through the network. The layer configurations for the Generator across the datasets are as follows:

- **SEED-V:** Layers with 64, 128, and 256 units.
- **DEAP:** Layers with 128 and 128 units.
- **GraffitiVR:** Layers with 64 and 64 units.

The final layer of the Generator is a linear transformation that maps the accumulated feature dimensions to the output dimension, producing synthetic EEG data intended to replicate real EEG signals.

**Discriminator architecture:** The Discriminator differentiates between real and synthetic EEG data. Each layer consists of a linear transformation followed by a LeakyReLU activation function with a negative slope coefficient of 0.2 and a dropout rate of 0.5 to prevent overfitting. The final layer outputs a single value for binary classification, which in some configurations passes through a sigmoid activation function to map it to a probability. The layer configurations for the Discriminator across the datasets are as follows:

- **SEED-V:** Layers with 128 and 64 units.
- **DEAP:** Layers with 64 and 64 units.
- **GraffitiVR:** Layers with 32 and 32 units.

**Gradient penalty implementation:** For the CWGAN-GP variant, a gradient penalty term is added to the Discriminator's loss function to stabilize training. The penalty is calculated by interpolating between real and generated data, computing the gradient of the Discriminator's output with respect to this interpolation, and enforcing the gradient norm to remain close to 1.

### Implementation details

The network structures and parameters are implemented using PyTorch's neural network module (`nn.Module`). In the Generator class, residual blocks iteratively expand the feature space before the final output layer. This architecture is crucial for capturing the complex patterns required to synthesize realistic EEG-like signals from non-EEG inputs.

The Discriminator employs LeakyReLU activations and dropout layers to improve training stability and prevent overfitting. The combination of these techniques enables the HATL framework to effectively learn the characteristics of real EEG signals across different datasets.

To evaluate the effectiveness of the HATL architecture, we compare the performance of different GAN variants in generating EEG data for emotion recognition tasks. Classifier performance is averaged across multiple models, including SVM, RF, and MLP, to mitigate overfitting risks and provide a generalized assessment of the system.

### Computational complexity considerations for HATL

The computational complexity of the proposed HATL framework varies based on the specific GAN architecture used for EEG data generation. CWGAN-GP introduces an

added complexity due to the gradient penalty computation, which increases training time compared to simpler GAN architectures like CGAN. The complexity of CWGAN-GP is approximately $O(N^2)$ per iteration, where $N$ is the number of parameters in the model. This contrasts with traditional emotion recognition models using real EEG data, which do not involve generative steps and are thus less computationally intensive.

Compared to baseline emotion recognition models that solely use real EEG data, the HATL framework requires additional resources for training the GAN. However, this upfront cost is offset by the reduced need for extensive calibration, particularly in applications like VR where calibration duration significantly impacts user experience.

In practice, the HATL approach remains feasible for real-time applications, as the synthetic EEG data generation can be precomputed and reused, thus not impacting runtime during classification. The improved calibration efficiency, combined with high accuracy, makes HATL a computationally efficient alternative for immersive environments, despite the initial training complexity.

## Evaluation criteria

### Data similarity metrics

To evaluate the similarity between real and generated EEG data, we employ three primary distance metrics—Euclidean distance, Wasserstein distance, and KL divergence—as well as statistical analysis through T-Test (T statistic and $p$-value).

$$\text{Euclidean}(U, V) = \sqrt{\sum_{i=1}^{N} (u_i - v_i)^2}. \qquad (5)$$

The Euclidean distance provides a measure of similarity by calculating the direct spatial difference between corresponding points in the real and generated EEG data. Given two vectors, $U = \{u_1, u_2, \ldots, u_N\}$ and $V = \{v_1, v_2, \ldots, v_N\}$, where $N$ is the number of points and $i$ is the index of each point, the Euclidean Distance is defined in Eq. (5). This metric captures the spatial alignment between real and synthetic EEG data.

The Wasserstein distance (also known as Earth Mover's distance) assesses the similarity between the distributions of real and generated EEG data. This metric takes into account the amount of "effort" required to transform one distribution into the other, making it suitable for evaluating how closely the overall patterns of synthetic data align with the distribution of real EEG data.

$$\text{KL}(P \parallel Q) = \sum_{i} P(i) \log \frac{P(i)}{Q(i)}. \qquad (6)$$

KL divergence quantifies the divergence between the probability distributions of generated and real EEG data. Given two distributions $P$ (real EEG) and $Q$ (generated EEG), the KL divergence is calculated in Eq. (6). Lower KL divergence values indicate a closer match between the distributions, suggesting that the model effectively captures the statistical properties of real EEG data.

The T-Test analysis includes the T statistic and *p*-value to statistically assess the difference between real and generated EEG data for each EEG band. The T statistic quantifies the magnitude of difference, while the *p*-value indicates the significance of this difference. A higher *p*-value (typically above 0.05) suggests that the generated and real data are not significantly different, indicating high fidelity in the generated EEG signals.

### Classification performance metrics

Accuracy is a widely utilized metric for evaluating the performance of classification models. In Eq. (7), *TP* represents the number of true positives, which are the instances correctly identified as positive by the model. *TN* denotes the number of true negatives, referring to the instances correctly identified as negative by the model. *FP* stands for the number of false positives, which are the instances incorrectly identified as positive. *FN* is the number of false negatives, referring to the instances incorrectly identified as negative. Accuracy, in this context, serves as a straightforward and intuitive measure of our model's overall effectiveness in correctly predicting emotional states. It is particularly valuable in scenarios where the dataset exhibits a balanced distribution across different classes or categories.

$$\text{Accuracy} = \frac{TP + TN}{TP + TN + FP + FN}. \tag{7}$$

In addition to accuracy, precision and recall are also crucial metrics for evaluating the performance of classification models, especially when dealing with imbalanced datasets. Precision, or positive predictive value, measures the accuracy of positive predictions. Recall, also known as sensitivity or true positive rate, measures the ability of a model to identify all relevant instances. Mathematically, precision and recall are defined in Eq. (8).

$$\text{Precision} = \frac{TP}{TP + FP} \qquad \text{Recall} = \frac{TP}{TP + FN}. \tag{8}$$

Additionally, the receiver operating characteristic (ROC) curve and the area under the curve (AUC) score are essential metrics for evaluating classification performance. The ROC curve is a graphical representation of a model's diagnostic ability across various threshold settings. It is plotted with the true positive rate on the y-axis and the false positive rate on the x-axis. The ROC curve illustrates the trade-off between sensitivity and specificity, providing insight into the model's performance at different classification thresholds. The area under the ROC curve (AUC) quantifies the overall ability of the model to distinguish between positive and negative classes. The AUC score ranges from 0 to 1, with higher values indicating better model performance.

These metrics provide a holistic framework for evaluating classification performance. Accuracy offers a straightforward measure of overall correctness, while precision and recall give insight into the model's handling of positive predictions, which is especially crucial for imbalanced datasets. ROC curves and AUC scores add a threshold-independent perspective, revealing the model's discriminative power across varying classification thresholds.

### Calibration improvement score

To additionally assess the improvement in calibration efficiency brought by our model, we introduce a novel metric, calibration improvement score (CIS), which measures the reduction in calibration time or steps needed to prepare the model for use with a new subject. The CIS is defined as the percentage decrease in calibration time or steps compared to a baseline model without the proposed enhancements. A higher CIS indicates a more significant improvement in calibration efficiency.

$$CIS = \left(1 - \frac{B}{A}\right) \times 100\%. \tag{9}$$

Given a baseline calibration duration of $A$ units (*e.g.*, seconds, minutes, or calibration steps) and our model's calibration duration of $B$ units, the CIS is defined in Eq. (9).

## Experimental framework

Table 3 summarizes the training, calibration, and testing data splits used for implementing the HATL architecture across the SEED-V, DEAP, and GraffitiVR datasets. Adopting a Leave-One-Subject-Out (LOSO) strategy, our experimental framework is carefully crafted to provide a detailed evaluation of the model's capabilities. In this setup, for every test subject, the GAN models are meticulously trained using the data from the other 15 training subjects (training phase in Fig. 1). This uniform method, applied across the SEED-V, DEAP, and GraffitiVR datasets, facilitates a comprehensive analysis of the GAN's capability in generating EEG-like signals with minimal inputs and its effectiveness in improving the precision of emotion recognition classifiers.

The emotion classification tasks in this study were designed as multiclass problems, reflecting the structure of the datasets. The SEED-V dataset includes five emotional class labels: Happy, Sad, Disgust, Neutral, and Fear, representing specific emotional states. For the DEAP and GraffitiVR datasets, emotional responses are categorized within the Valence-Arousal framework, divided into four quadrants: HVHA, HVLA, LVLA, and LVHA. These multiclass classification tasks allow for a nuanced evaluation of emotional states across datasets.

The SEED-V dataset includes three recording sessions for each subject. For the test subject's data were partitioned into two subsets: two sessions for calibration (calibration phase in Fig. 1) and one for testing (testing phase in Fig. 1). In the DEAP dataset, test subject's data was split into two portions: one part for calibration (calibration phase in Fig. 1) and one for testing (testing phase in Fig. 1). In the GraffitiVR dataset, test subject's data divided into an 80–20% split, 80% for calibration (calibration phase in Fig. 1) and 20% for testing (testing phase in Fig. 1).

In each case, we compared the performance of different GAN variations, specifically CGAN, CWGAN, and CWGAN-GP. To provide a comprehensive evaluation of multiclass classification tasks, the performance of multiple classifiers (SVM, RF, MLP) was averaged. This approach mitigated biases towards any single classifier and offered a more balanced assessment of the HATL architecture's capabilities.

**Table 3 Details of training, calibration, and testing data splits.**

| Dataset | Training (Training subjects) | Calibration (Test subject) | Testing (Test subject) |
|---|---|---|---|
| SEED-V | Data from 15 training subjects | Data from sessions #1 and #2 for 1 test subject (LOSO) | Data from session #3 for the same test subject |
| DEAP | Data from 15 training subjects | One part of data for 1 test subject (LOSO) | The remaining part of data for the same test subject |
| GraffitiVR | Data from 15 training subjects | 80% of data for 1 test subject (LOSO) | 20% of data for the same test subject |

# RESULTS

In this section, we present an in-depth evaluation of our approach to generating synthetic EEG data using CGAN, CWGAN, and CWGAN-GP models. We begin by examining how well the generated data represents real EEG from both statistical and neuroscientific perspectives, comparing generated and real EEG data across multiple datasets using metrics such as Euclidean and Wasserstein distances. Next, we explore examples of EEG mappings to illustrate alignment between real and synthetic data across frequency bands. We then present findings on emotion recognition performance, comparing classifiers trained on synthetic EEG, real EEG, and non-EEG sensory data across different datasets. Finally, we assess the calibration improvement performance, focusing on how calibration duration can be optimized using synthetic EEG data in immersive VR environments.

## Performance of EEG data generation

To assess the fidelity of synthetic EEG data generated by the CGAN, CWGAN, and CWGAN-GP models, we compared generated samples to real samples on a row-by-row basis. Each row in the datasets represents EEG band values across channels, corresponding to a specific emotional response as labeled in the dataset. Table 4 shows the Euclidean and Wasserstein distances across the SEED-V (*Liu et al., 2021*), DEAP (*Koelstra et al., 2011*), and GraffitiVR (*Karakas et al., 2024*) datasets. The Euclidean distance captures the spatial similarity between real and generated EEG samples, indicating how closely the generated data mirrors individual real samples. In contrast, the Wasserstein distance assesses the alignment of the data distributions, with lower values suggesting a closer match between the distribution of generated and real EEG data.

In the SEED-V dataset, CWGAN-GP achieved the lowest Euclidean and Wasserstein distances, highlighting its ability to closely approximate the spatial and distributional features of real EEG data. The slight reduction in Wasserstein distance for CWGAN-GP compared to other models underscores its effectiveness in capturing the underlying distributional characteristics of EEG signals.

For the DEAP dataset, CWGAN-GP also exhibited the lowest Euclidean and Wasserstein distances among all models, demonstrating strong alignment with real EEG patterns. This result underscores CWGAN-GP's capacity to preserve both spatial coherence and distributional fidelity, which are essential for maintaining the integrity of EEG data across diverse datasets.

**Table 4 Computed distances using CGAN, CWGAN, and CWGAN-GP across different datasets.**
Bold texts indicate the lowest distances achieved for each dataset and metric.

| Dataset | Distance metrics | CGAN | CWGAN | CWGAN-GP |
|---|---|---|---|---|
| SEED-V | Euclidean | 12.33 ± 3.49 | 11.8 ± 3.46 | **10.27 ± 2.83** |
| | Wasserstein | 0.31 ± 0.20 | 0.30 ± 0.20 | **0.27 ± 0.16** |
| DEAP | Euclidean | 5.15 ± 1.05 | 4.64 ± 1.05 | **4.50 ± 1.07** |
| | Wasserstein | 0.20 ± 0.12 | 0.19 ± 0.12 | **0.17 ± 0.12** |
| GraffitiVR | Euclidean | 4.31 ± 2.45 | 3.81 ± 2.51 | **3.79 ± 2.30** |
| | Wasserstein | 0.49 ± 0.41 | 0.46 ± 0.44 | **0.43 ± 0.41** |

In the GraffitiVR dataset, CWGAN-GP again achieved Euclidean and Wasserstein distances comparable to real EEG samples, suggesting that the model effectively captures the nuanced characteristics of EEG data in immersive VR settings.

Overall, CWGAN-GP consistently shows the highest similarity to real EEG across all datasets, as evidenced by its lower Euclidean and Wasserstein distances. These findings underscore the reliability of CWGAN-GP in generating synthetic EEG data that closely mirrors the spatial and distributional features of authentic EEG signals.

Figure 2 illustrates the progression of Euclidean and Wasserstein distances across training epochs for the DEAP, SEED-V, and GraffitiVR datasets, providing a comparative view of real EEG data and synthetic data generated by the CWGAN-GP model. Lower distances indicate higher similarity to real EEG patterns and distributions. This visualization demonstrates how the similarity between generated and real EEG data improves over time, with decreasing distances indicating better alignment in both temporal and distributional properties.

The observed convergence in distance metrics by the 100th epoch suggests that this training duration is adequate for the CWGAN-GP model to capture essential EEG features. Notably, after approximately the 70th epoch, both Euclidean and Wasserstein distances stabilize at lower values, signaling that the model has effectively learned the underlying patterns and distributional characteristics of real EEG data. Extending training beyond 100 epochs may offer limited additional benefit and could risk overfitting, where the model becomes overly attuned to the training data's noise or specificities. Thus, capping training at 100 epochs strikes an optimal balance between computational efficiency and the generation of high-quality, realistic EEG data.

Table 5 summarizes the performance of the CWGAN-GP model in generating synthetic EEG data across different EEG bands (Delta, Theta, Alpha, Beta, and Gamma) for the SEED-V (*Liu et al., 2021*), DEAP (*Koelstra et al., 2011*), and GraffitiVR (*Karakas et al., 2024*) datasets. The results highlight the similarity between generated and real EEG data in terms of KL Divergence, Wasserstein Distance, and T-Test analysis.

For the SEED-V dataset, the CWGAN-GP model demonstrates close alignment with real EEG data across most bands, with particularly low Wasserstein Distance values in the Gamma and Theta bands. This indicates strong distributional alignment, as reflected by the non-significant T-Test $p$-values in these bands.

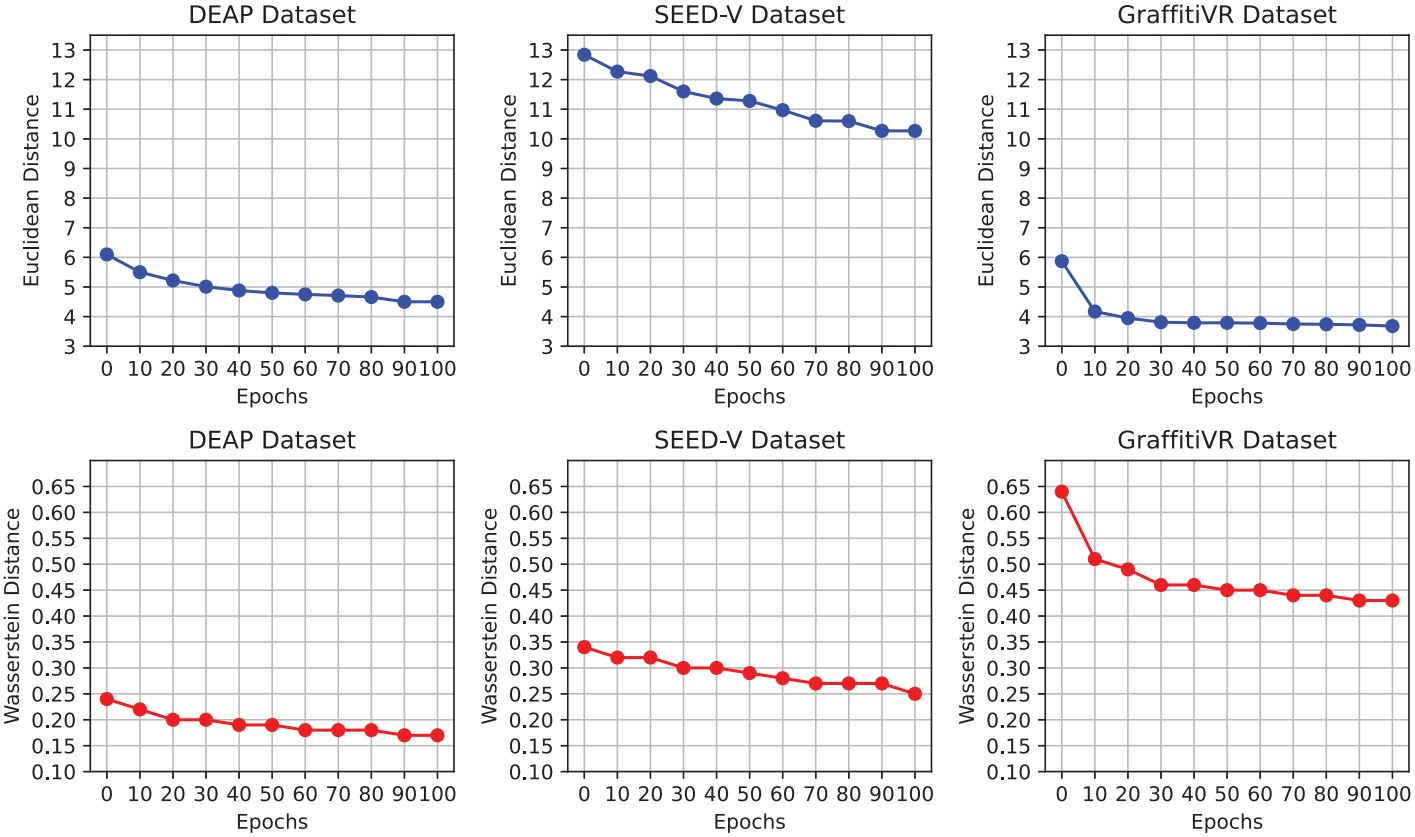

**Figure 2** Euclidean and Wasserstein distances across epochs for the DEAP, SEED-V, and GraffitiVR datasets, comparing real EEG data with synthetic EEG data generated by the CWGAN-GP model.

In the DEAP dataset, the model continues to show robust performance, with low KL Divergence and Wasserstein Distance values in the Theta and Alpha bands, suggesting accurate representation of real EEG characteristics. The T-test results support this observation, showing minimal statistically significant differences between generated and real data in these bands.

The GraffitiVR dataset results further underscore the model's consistency, with low distance metrics across all EEG bands and non-significant t-Test results in most cases. The Theta band, however, presents a slightly higher T-Statistic, though the *p*-value remains within acceptable limits, indicating that the generated data still closely approximates real EEG signals.

Overall, these results demonstrate that the CWGAN-GP model effectively replicates the statistical and distributional properties of real EEG data across multiple frequency bands and datasets. The model's performance is consistent across the SEED-V, DEAP, and GraffitiVR datasets, confirming its suitability for generating synthetic EEG data that mirrors real EEG characteristics in a reliable and generalizable manner.

To further evaluate the quality of the generated EEG data, we present examples with high error and low error of EEG mappings for each dataset: SEED-V, DEAP, and

**Table 5 Comparison of KL divergence, Wasserstein Distance, and T-Test results between CWGAN-GP generated EEG and real EEG across EEG bands for SEED-V, DEAP, and GraffitiVR datasets.** Bold $p$-values indicate non-significant results ($p >= 0.05$).

| Dataset | EEG bands | KL divergence | Wasserstein distance | T statistic | T $p$-value |
|---------|-----------|---------------|----------------------|-------------|-------------|
| SEED-V | Delta | 4.08 | 0.04 | 1.00 | **0.32** |
| | Theta | 1.94 | 0.02 | −0.29 | **0.77** |
| | Alpha | 0.82 | 0.03 | 0.71 | **0.48** |
| | Beta | 1.52 | 0.03 | −1.42 | **0.16** |
| | Gamma | 0.77 | 0.02 | 0.03 | **0.98** |
| DEAP | Theta | 0.88 | 0.02 | −0.55 | **0.58** |
| | Alpha | 1.11 | 0.02 | 0.42 | **0.68** |
| | Beta | 0.29 | 0.02 | −1.40 | **0.16** |
| | Gamma | 4.97 | 0.05 | −0.21 | **0.84** |
| GraffitiVR | Delta | 0.66 | 0.03 | −1.34 | **0.18** |
| | Theta | 0.82 | 0.07 | −2.31 | 0.02 |
| | Alpha | 1.06 | 0.07 | −0.51 | **0.61** |
| | Beta | 1.58 | 0.09 | −0.52 | **0.60** |
| | Gamma | 0.95 | 0.08 | 0.40 | **0.69** |

GraffitiVR. Examples with low error have lower Euclidean and Wasserstein distances, indicating closer alignment with real EEG signals, while examples with high error have higher distances, showing greater discrepancies. Each dataset varies in terms of EEG channel placement and band representations, which impacts the mapping and alignment between real and generated data.

Each row in the datasets represents EEG band values across channels, corresponding to a specific emotional response as labeled in the dataset. The color code in the Figs. 3–8 visualizes example rows of these EEG feature values, with each figure representing a single row tied to a specific emotional label. To enhance clarity, the EEG data from all three datasets was normalized to the range of (−1,1) which helps provide better visualizations and demonstrate proper alignments. The Python MNE package was used to generate the EEG brain map visualizations. Warm colors (*e.g.*, red) indicate higher activity levels in EEG bands, while cool colors (*e.g.*, blue) indicate lower activity levels, with darker shades reflecting values closer to the extremes of the range. These visualizations offer an intuitive representation of the EEG features.

The SEED-V dataset includes 62 EEG channels and captures five frequency bands: delta, theta, alpha, beta, and gamma. Figures 3 and 4 display examples with low error and high error, respectively, where generated EEG signals are compared against real EEG data across these bands. In the example with low error (Fig. 3, corresponding to the Fear emotional label), the generated data shows close alignment to real EEG signals, especially in the theta and alpha bands. The example with high error (Fig. 4, corresponding to the Disgust emotional label), however, demonstrates higher divergence, with noticeable deviations in the beta and gamma bands.
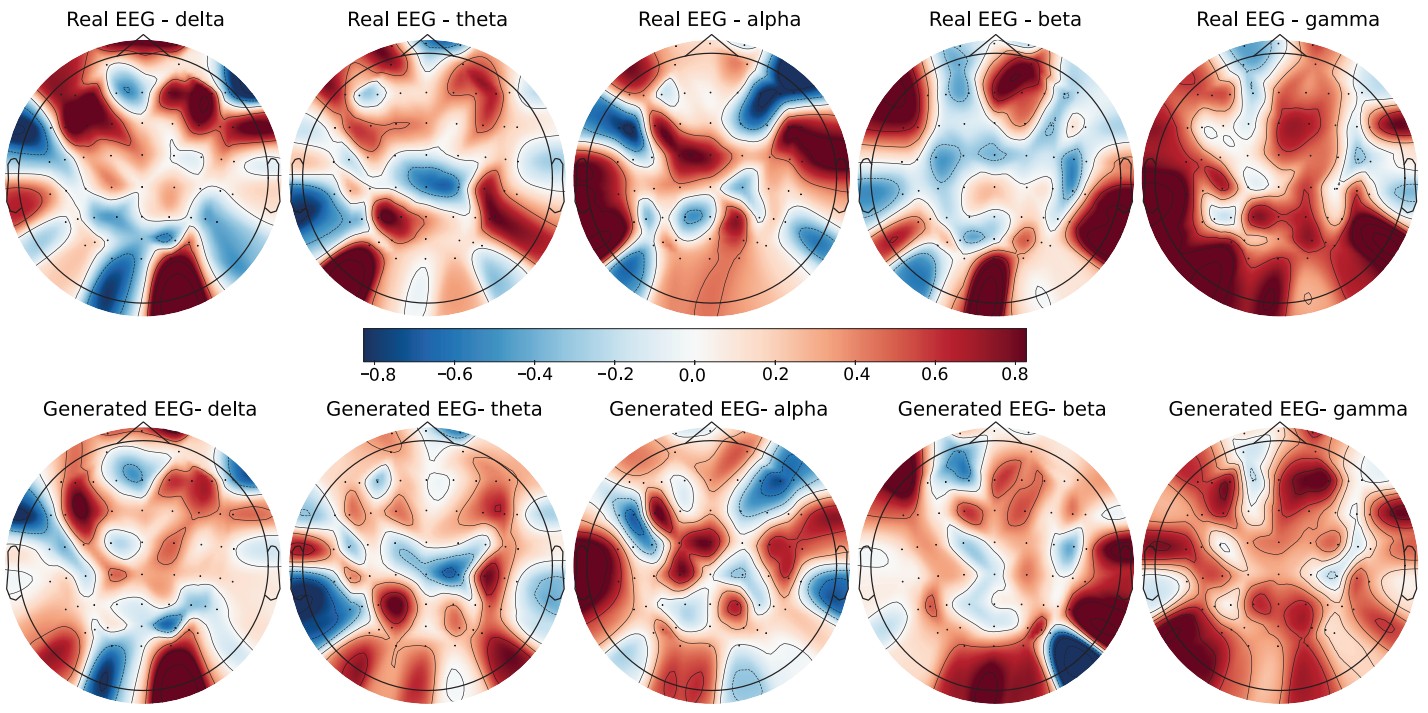

**Figure 3** Real EEG and generated EEG with low error for the SEED-V dataset.

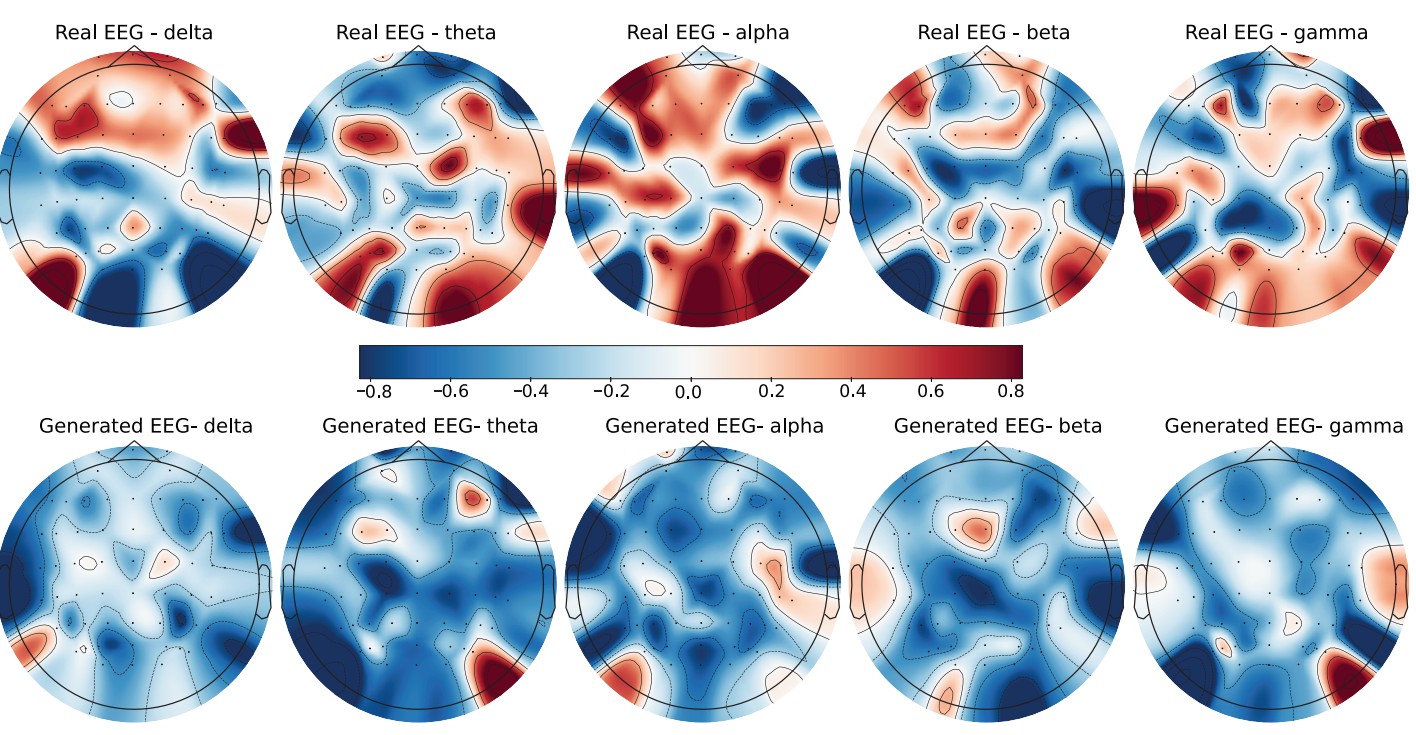

**Figure 4** Real EEG and generated EEG with high error for the SEED-V dataset.

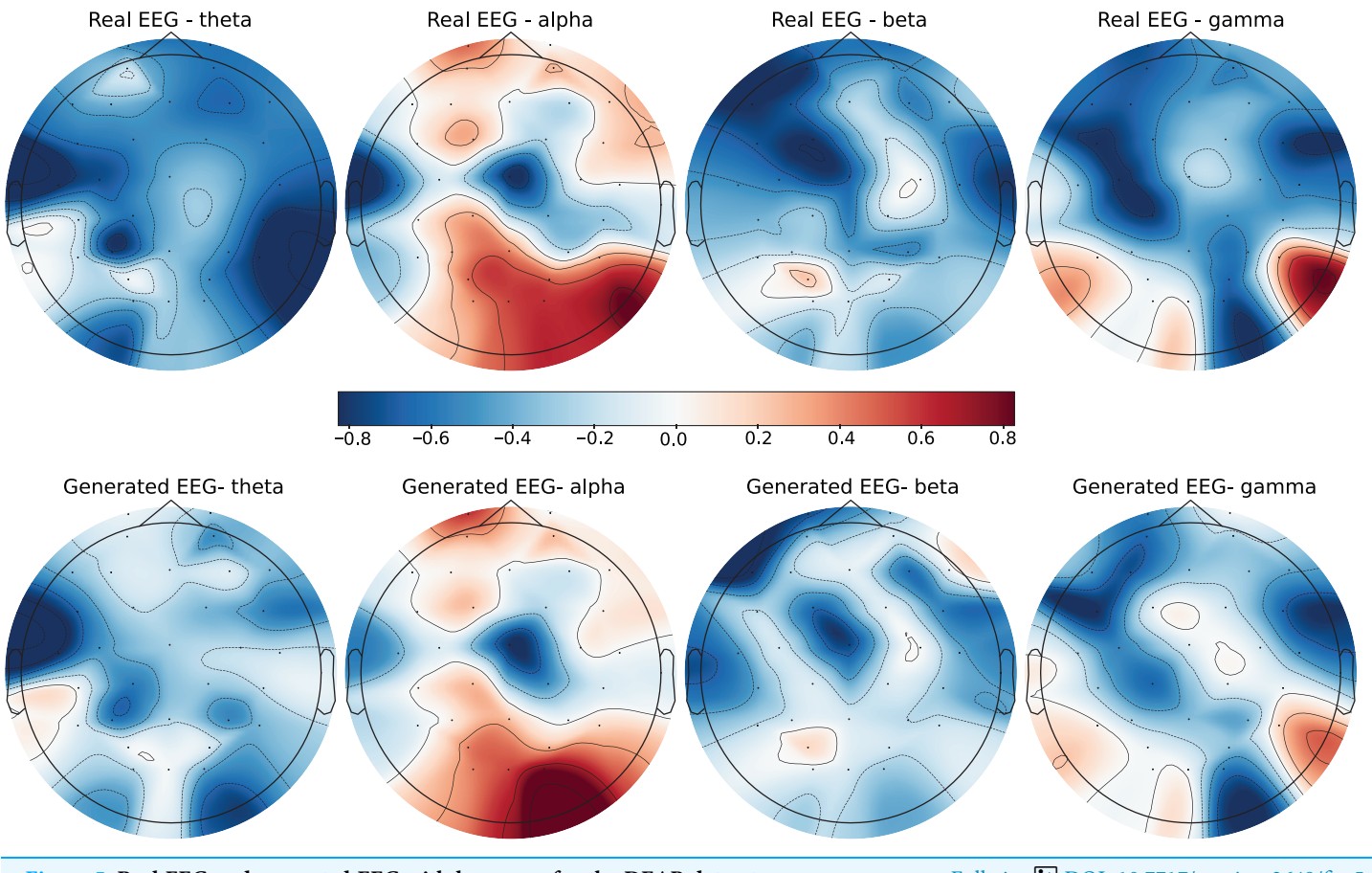

**Figure 5  Real EEG and generated EEG with low error for the DEAP dataset.**

The DEAP dataset contains 32 EEG channels and primarily focuses on four frequency bands: theta, alpha, beta, and gamma. Figures 5 and 6 illustrate examples with low error and high error, respectively. In the example with low error (Fig. 5, corresponding to the HVHA emotional label), the generated EEG shows high similarity to real EEG, particularly in the beta and gamma bands, which are essential for emotional response modeling. Conversely, the example with high error (Fig. 6, corresponding to the LVLA emotional label) reveals a poor alignment in the alpha and gamma bands, suggesting limitations in the generative model's ability to accurately replicate real data for this subject.

The GraffitiVR dataset utilizes six EEG channels and also records five frequency bands: delta, theta, alpha, beta, and gamma. Figures 7 and 8 showcase examples with low error and high error, respectively. In the example with low error (Fig. 7, corresponding to the HVLA emotional label), the generated EEG aligns closely with real EEG in the delta and beta bands, which are significant in VR-based affective assessments. The example with high error (Fig. 8, corresponding to the LVLA emotional label), however, shows considerable misalignment across most bands, indicating challenges in capturing EEG characteristics accurately within VR contexts.

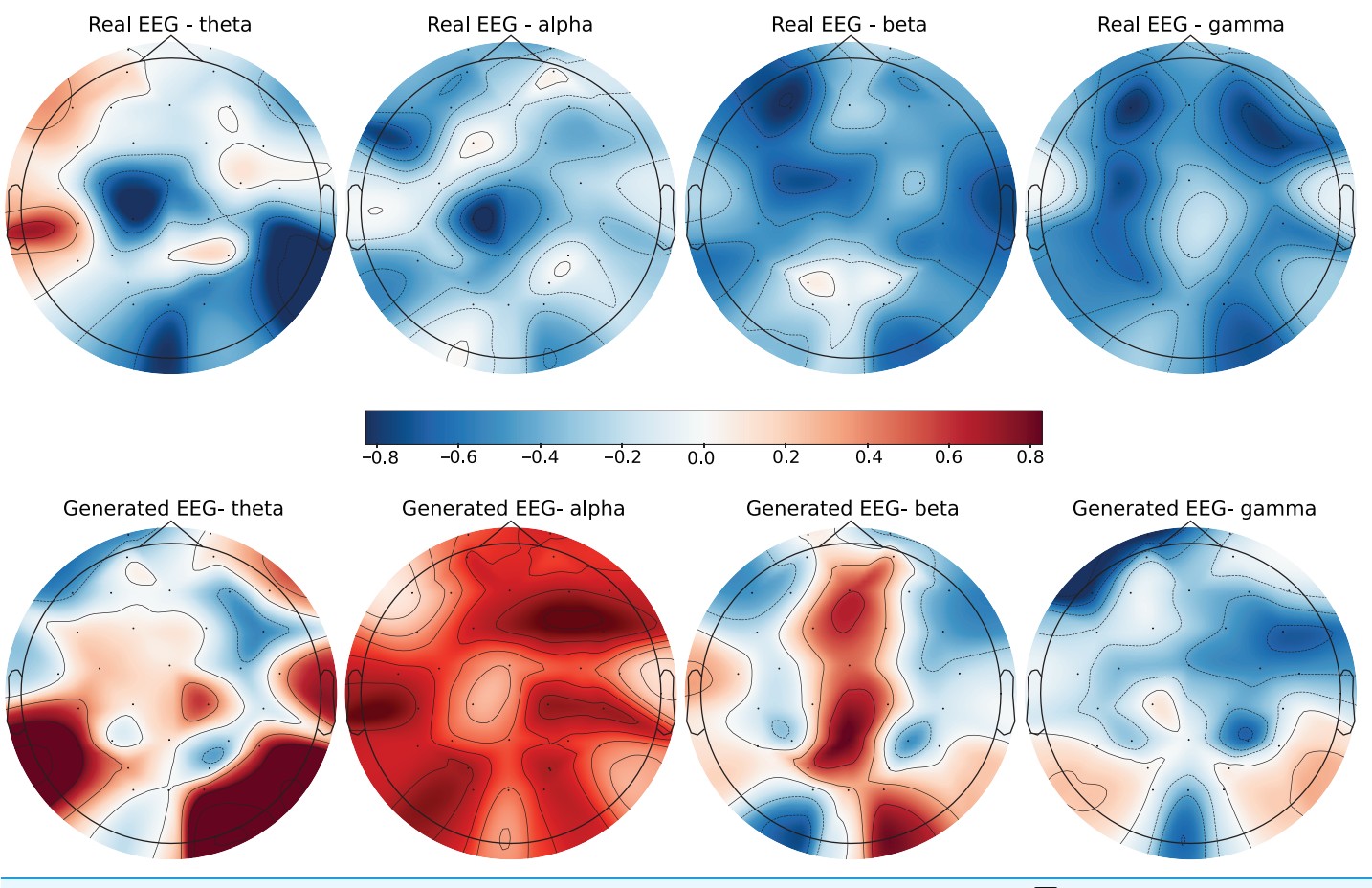

**Figure 6** **Real EEG and generated EEG with high error for the DEAP dataset.**

Across the datasets, the examples with low error show strong alignment between real and generated EEG data, particularly in the lower Euclidean and Wasserstein distance bands such as alpha and beta. This alignment suggests that CWGAN-GP effectively captures the temporal dynamics and frequency-specific features of EEG data. In contrast, the examples with high error exhibit inconsistencies, primarily in the gamma and beta bands, where the generative model struggles to replicate the complexities of real EEG signals. These comparisons underscore the importance of frequency-specific tuning in EEG synthesis for applications in emotion recognition and VR.

## Emotion recognition results using different GAN models

To understand the key elements that influence the efficacy of emotion recognition, we compared GANs on the SEED-V (*Liu et al., 2021*), DEAP (*Koelstra et al., 2011*), and GraffitiVR (*Karakas et al., 2024*) datasets. These studies aimed to assess the results of the proposed model, focusing on the performance similarity between the original data and the synthetic EEG data when combined with non-EEG sensory data. The experiments benchmarked the performance of three GAN models: CGAN, CWGAN, and CWGAN-

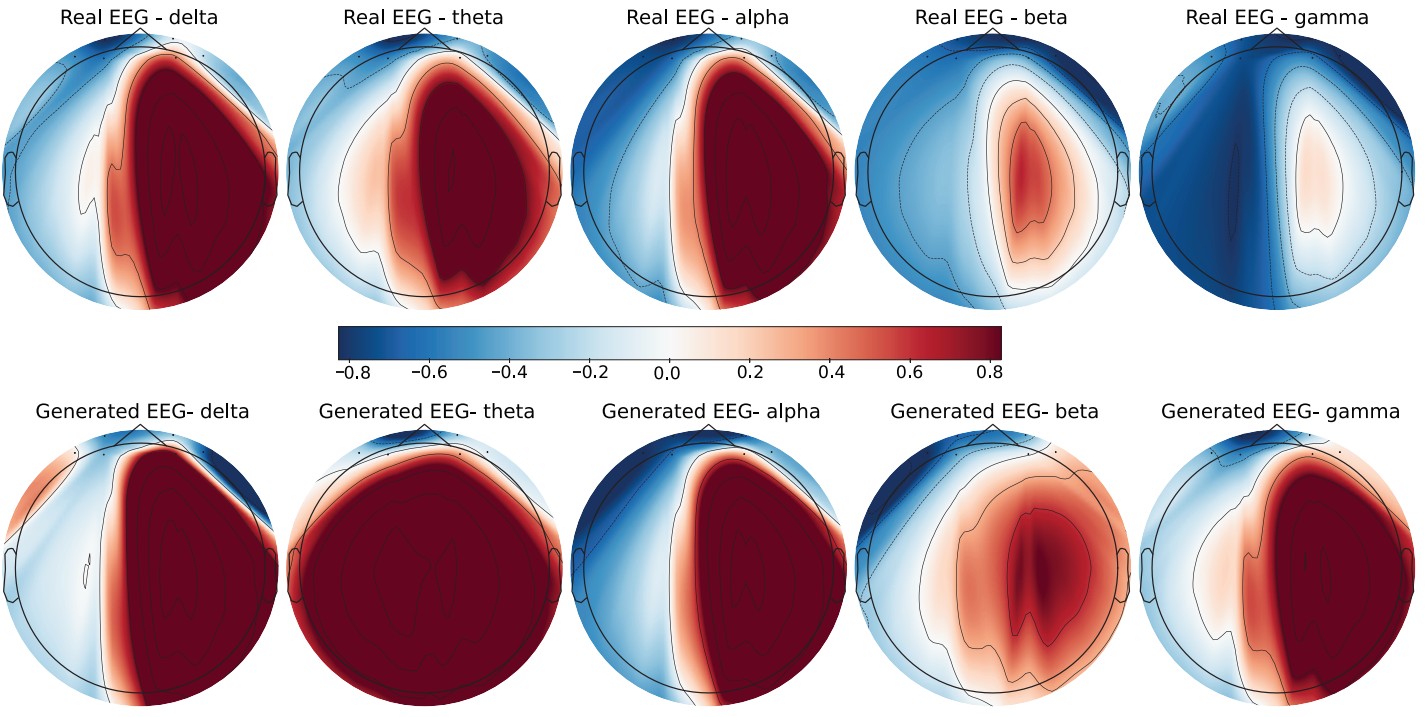

**Figure 7** Real EEG and generated EEG with low error for the GraffitiVR dataset.

**Figure 8** Real EEG and generated EEG with high error for the GraffitiVR dataset.

**Table 6 Performance comparison results.**

|  | Only Non-EEG sensory data | CGAN | CWGAN | CWGAN-GP | Real EEG, Non-EEG sensory data |
|---|---|---|---|---|---|
| **SEED-V** |  |  |  |  |  |
| Mean accuracy | 0.50 ± 0.06 | 0.54 ± 0.08 | 0.56 ± 0.10 | 0.59 ± 0.09 | 0.63 ± 0.10 |
| Mean precision | 0.50 ± 0.07 | 0.54 ± 0.09 | 0.56 ± 0.10 | 0.60 ± 0.09 | 0.63 ± 0.10 |
| Mean recall | 0.51 ± 0.07 | 0.55 ± 0.09 | 0.58 ± 0.10 | 0.60 ± 0.09 | 0.64 ± 0.11 |
| SVM accuracy | 0.50 ± 0.09 | 0.54 ± 0.09 | 0.56 ± 0.10 | 0.59 ± 0.10 | 0.64 ± 0.11 |
| RF accuracy | 0.51 ± 0.07 | 0.54 ± 0.10 | 0.56 ± 0.11 | 0.59 ± 0.06 | 0.67 ± 0.11 |
| MLP accuracy | 0.49 ± 0.06 | 0.53 ± 0.08 | 0.56 ± 0.09 | 0.58 ± 0.08 | 0.59 ± 0.11 |
| **DEAP** |  |  |  |  |  |
| Mean accuracy | 0.48 ± 0.10 | 0.49 ± 0.11 | 0.50 ± 0.10 | 0.51 ± 0.11 | 0.51 ± 0.12 |
| Mean precision | 0.48 ± 0.10 | 0.49 ± 0.11 | 0.50 ± 0.11 | 0.51 ± 0.12 | 0.51 ± 0.12 |
| Mean recall | 0.49 ± 0.11 | 0.50 ± 0.11 | 0.51 ± 0.11 | 0.52 ± 0.11 | 0.52 ± 0.12 |
| SVM accuracy | 0.49 ± 0.12 | 0.50 ± 0.13 | 0.51 ± 0.12 | 0.52 ± 0.13 | 0.52 ± 0.13 |
| RF accuracy | 0.52 ± 0.10 | 0.55 ± 0.11 | 0.55 ± 0.11 | 0.57 ± 0.12 | 0.54 ± 0.11 |
| MLP accuracy | 0.43 ± 0.09 | 0.42 ± 0.12 | 0.44 ± 0.12 | 0.44 ± 0.11 | 0.47 ± 0.11 |
| **GraffitiVR** |  |  |  |  |  |
| Mean accuracy | 0.66 ± 0.14 | 0.68 ± 0.13 | 0.70 ± 0.14 | 0.72 ± 0.13 | 0.73 ± 0.14 |
| Mean precision | 0.66 ± 0.14 | 0.69 ± 0.14 | 0.70 ± 0.14 | 0.72 ± 0.13 | 0.73 ± 0.14 |
| Mean recall | 0.67 ± 0.15 | 0.69 ± 0.14 | 0.71 ± 0.15 | 0.73 ± 0.14 | 0.74 ± 0.15 |
| SVM accuracy | 0.66 ± 0.14 | 0.68 ± 0.13 | 0.70 ± 0.12 | 0.72 ± 0.12 | 0.73 ± 0.14 |
| RF accuracy | 0.68 ± 0.12 | 0.70 ± 0.13 | 0.72 ± 0.12 | 0.74 ± 0.12 | 0.76 ± 0.13 |
| MLP accuracy | 0.64 ± 0.11 | 0.66 ± 0.11 | 0.68 ± 0.12 | 0.70 ± 0.10 | 0.70 ± 0.11 |

GP, as well as configurations using real EEG data and solely non-EEG sensory data. The summarized results, as illustrated in Table 6, highlight the relative performances across different methods.

The precision and recall metrics across the different setups closely mirror the accuracy rates, indicating that the datasets used were balanced with respect to each emotional label. This balance in the dataset ensures that the performance metrics are not skewed by a disproportionate representation of any class, providing a reliable measure of model performance across different emotional states.

In comparing individual classifier performances, the RF classifier consistently shows superior accuracy across all datasets and configurations, particularly excelling when real EEG and non-EEG sensory data are combined. This suggests that the RF model's ability to handle heterogeneous and complex data structures is more effective, particularly in environments rich in multimodal data. The SVM and MLP also show commendable performances, but the RF's ensemble method, which integrates multiple decision trees, appears to better capture the nuances in varied data types, leading to more robust classification. Given the varied performance across SVM, RF, and MLP classifiers, averaging the results of these three models provides a balanced evaluation metric that mitigates individual model biases and capitalizes on their collective strengths. This

approach ensures a fair and comprehensive assessment of GAN models' efficacy in emotion recognition, thereby supporting more generalized and reliable conclusions.

For the SEED-V dataset, there's a notable difference in emotion recognition classifiers' accuracy between models that include EEG data and those that rely solely on non-EEG sensory data. This difference is more significant compared to other datasets, indicating that SEED-V's structure and composition, where EEG and the non-EEG sensory data possibly present a distinct mapping to emotional states, thus allowing for greater differentiation in model performance. Specifically, the CWGAN-GP model, which integrates Generated EEG with the non-EEG sensory data, shows higher accuracy (0.59) on SEED-V, demonstrating its superior capability in capturing and synthesizing the nuanced interplay between physiological signals and emotional states within this dataset.

In the DEAP dataset, the accuracy metrics across various models are quite similar, indicating that the mapping between non-EEG sensory data and EEG data might not differ significantly across models like CWGAN (0.50) and CWGAN-GP (0.51). This outcome could reflect DEAP's inherent characteristics, where the relationship between emotional states and physiological responses may be more subtly represented in non-EEG sensory data. This subtle representation makes it challenging for models to differentiate based solely on the given data, leading to closely aligned accuracy metrics across various approaches.

The study was extended to include our proprietary GraffitiVR dataset, which analyzes emotional and behavioral responses to urban graffiti in a VR setting. These findings demonstrate a consistent pattern of increasing accuracy from generated EEG configurations to real EEG data, highlighting the progressive improvement in emotion recognition as the models advanced in complexity. The CWGAN-GP model achieved the highest accuracy (0.72) among GAN-based approaches, with real EEG and non-EEG sensory data achieving slightly higher accuracy (0.73). This indicates that CWGAN-GP is effective in synthesizing EEG data that closely mimics real physiological signals in complex emotional recognition tasks. Additionally, the real EEG setup outperformed the GAN-generated configurations, emphasizing the importance of authentic EEG signals in capturing the nuances of human emotions, especially in immersive VR environments. The non-EEG sensory data configuration, while the least accurate, still provides valuable baseline information about emotional states. However, integrating EEG data, whether real or synthetic, significantly boosts accuracy, emphasizing the complementary nature of these signals in emotion recognition tasks. These findings validate the effectiveness of the proposed heterogeneous transfer learning approach in a VR context and suggest that advanced GAN techniques can enhance emotion recognition in immersive environments.

The ROC curves in Fig. 9 were generated by calculating the FPR and TPR for each class and then averaging these rates across all classes to produce a single, representative curve. This approach ensures a balanced assessment of the models' classification performance across different emotional states, as it mitigates the influence of class imbalances. The AUC values shown in the figure reflect this averaged ROC curve, offering a comprehensive

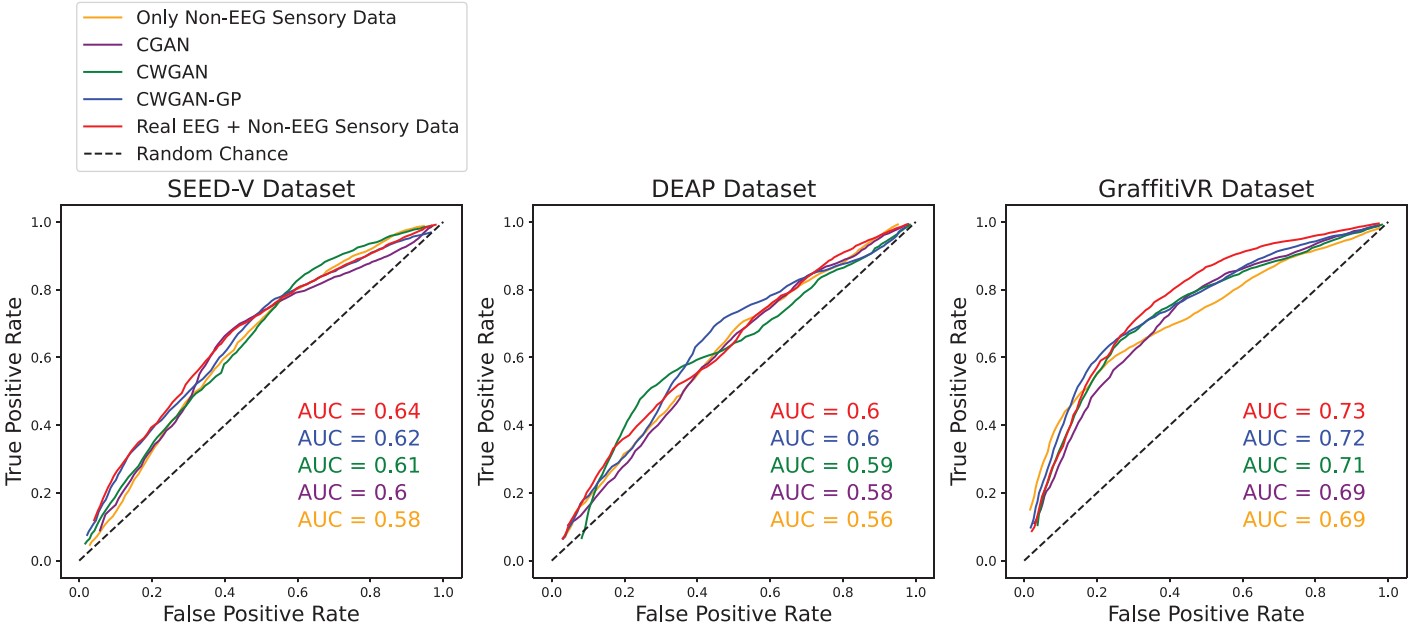

**Figure 9 ROC curves for emotion recognition across SEED-V, DEAP, and GraffitiVR datasets, comparing performance of different GAN-based models (CGAN, CWGAN, CWGAN-GP) and real EEG combined with non-EEG sensory data.**

measure of each model's ability to distinguish between emotional states. This method provides a robust evaluation metric for comparing the effectiveness of GAN-based models and real EEG data in emotion recognition tasks.

The ROC curves in Fig. 9 provide a detailed comparison of the emotion recognition performance across the SEED-V, DEAP, and GraffitiVR datasets. The CWGAN-GP model exhibits higher AUC values than the CGAN and CWGAN models, indicating superior accuracy in distinguishing between emotional states. Notably, the real EEG data combined with non-EEG sensory data achieves the highest AUC scores, reinforcing the significance of authentic EEG signals in emotion recognition. This trend highlights the effectiveness of CWGAN-GP in generating synthetic EEG that closely resembles real data, contributing positively to the overall emotion recognition performance in complex VR and multimodal settings.

## Emotion recognition results using SEED-V dataset

In Fig. 10, each bar shows the average accuracy of emotion recognition classifiers (SVM, RF, MLP), with black error bars indicating the standard deviation for individual classifiers. The y-axis shows accuracy, the x-axis shows subject IDs, and the last bar with 'mean' on the x-axis represents the average of all subjects.

The orange bar represents the accuracy for classifiers trained with non-EEG sensory data. The blue bar represents the accuracy for classifiers trained with

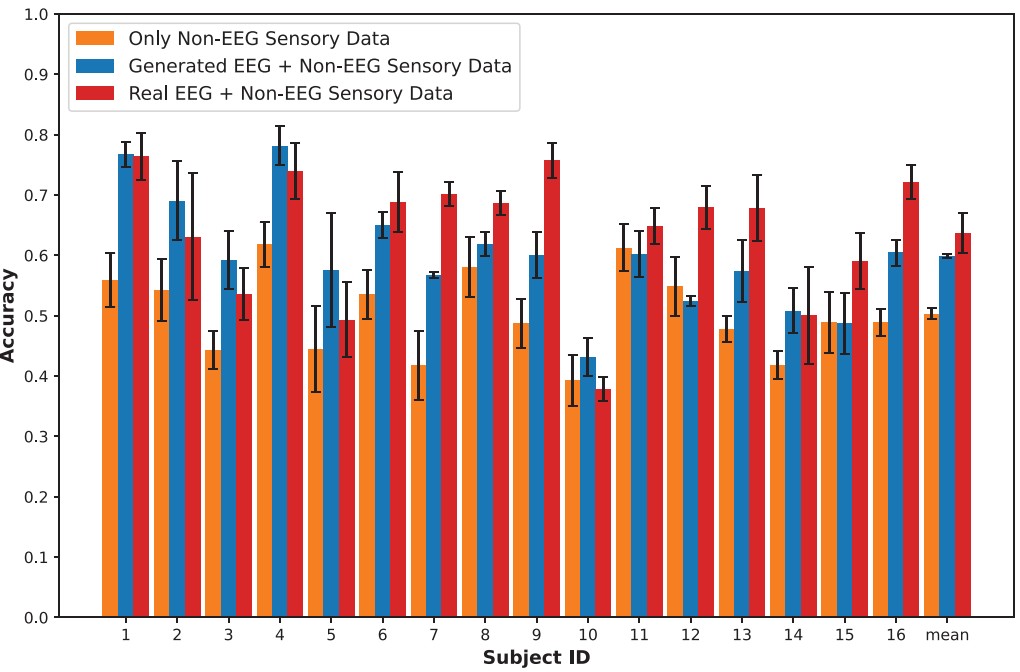

**Figure 10 CWGAN-GP results on SEED-V.**

CWGAN-GP-generated EEG and non-EEG sensory data. The red bar represents the accuracy for classifiers trained with real EEG and non-EEG sensory data.

The mean accuracy achieved with only non-EEG sensory data is 0.50. The mean accuracy with generated EEG and non-EEG sensory data is 0.59, demonstrating the potential of GAN models to significantly enhance classification accuracy beyond what is possible with only non-EEG sensory data. The mean accuracy with real EEG and non-EEG sensory data is 0.63.

The performance of classifiers trained with non-EEG sensory data is 79% compared to those trained with real EEG and non-EEG sensory data. Particularly noteworthy is the performance of generated EEG, which achieves 93% of the accuracy of real EEG and non-EEG sensory data, based on a calculation of 0.59/0.63.

This remarkable level of performance, nearly mirroring that of real EEG, underscores the viability of GAN-generated EEG signals as an effective alternative to real EEG data, especially in contexts where the latter is unavailable.

For individual subjects, such as Subject 1 and Subject 4, the accuracies achieved with generated EEG and non-EEG sensory data were exceptionally high, reaching 0.76 and 0.78, respectively. These figures represent better performance compared to real EEG and non-EEG sensory data, indicating that CWGAN-GP can produce noise-free signals that may even surpass the real ones.

This comprehensive analysis not only highlights the advantages of integrating EEG data, real or generated, with the non-EEG sensory data but also demonstrates the GAN's capability in generating EEG-like signals that closely mimic the accuracy of real EEG data in emotion recognition tasks.

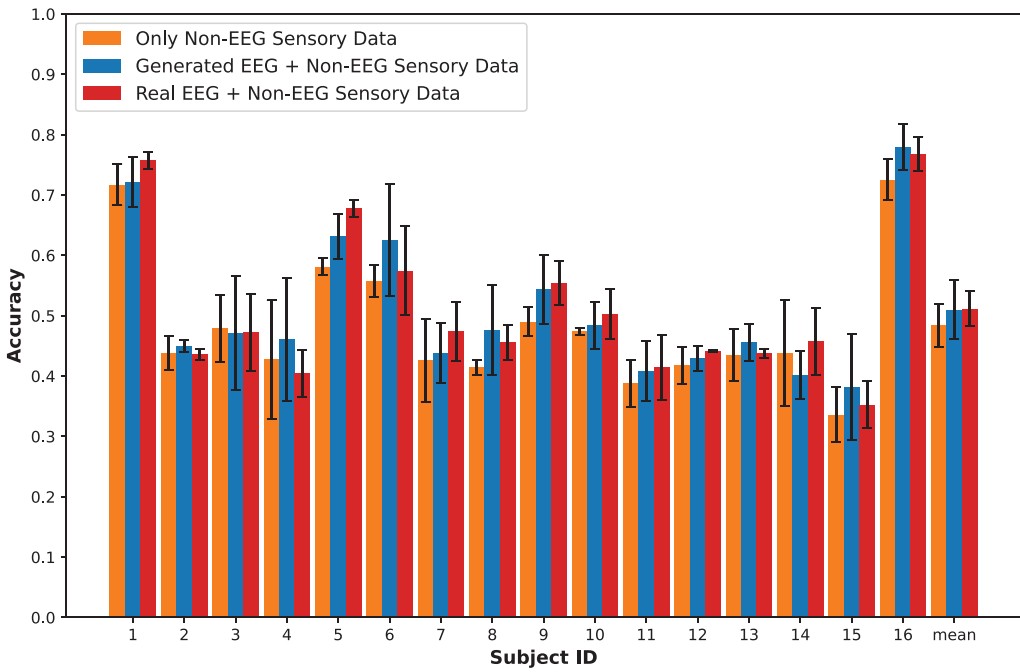

**Figure 11 CWGAN-GP results on DEAP.**

## Emotion recognition results using DEAP dataset

In Fig. 11, each bar and color represent the same aspects as in the SEED-V dataset. The DEAP dataset exhibits a unique phenomenon, where the accuracy metrics are notably close to each other. This pattern could be due to the intrinsic characteristics of the DEAP dataset, where emotional states might be more accurately reflected in the non-EEG sensory data. Consequently, the non-EEG sensory data closely mirror the information content provided by the EEG data regarding emotional state recognition.

The mean accuracy achieved with only non-EEG sensory data is 0.48. The mean accuracy achieved with generated EEG data combined with non-EEG sensory data is 0.51, which is nearly the same as the mean accuracy with real EEG data combined with non-EEG sensory data.

The performance of classifiers trained with non-EEG sensory data is 94% of those trained with real EEG and non-EEG sensory data. Particularly noteworthy is the performance of generated EEG, which achieves 99% of the accuracy of classifiers trained with real EEG and non-EEG sensory data.

This remarkable level of performance, nearly mirroring that of real EEG, underscores the potential of GAN-generated EEG signals. In some cases, such as with Subjects 2, 4, 6, 8, 13, 15, and 16, classifiers using generated EEG have better accuracy than those using real EEG.

A particularly exceptional case is Subject 16, where the accuracy with generated EEG and non-EEG sensory data is 0.77, surpassing the 0.76 accuracy achieved with real EEG and significantly outperforming the 0.72 accuracy with only the non-EEG sensory data. This instance highlights the potential of synthesized EEG signals in certain contexts to

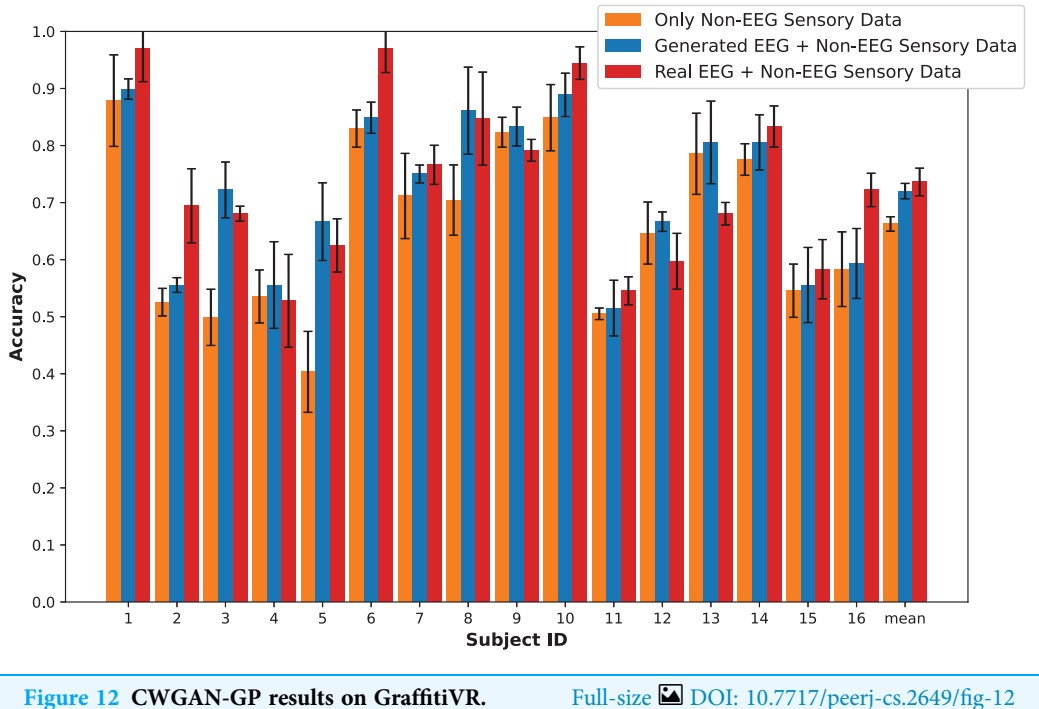

**Figure 12 CWGAN-GP results on GraffitiVR.**  

effectively complement or even substitute real EEG data for emotion recognition tasks within the DEAP dataset.

## Emotion recognition results using GraffitiVR dataset

In Fig. 12, each bar and its colors align with those in the SEED-V and DEAP datasets. Classifiers trained with only non-EEG sensory data yielded a mean accuracy of 0.66, while the use of generated EEG data combined with non-EEG sensory data raised the mean accuracy to 0.72. The mean accuracy achieved with real EEG and non-EEG sensory data was 0.73, indicating that classifiers trained with only non-EEG sensory data perform at 90% of the accuracy obtained with real EEG and non-EEG sensory data. A noteworthy finding is that the performance of classifiers trained with generated EEG achieved 97% of the accuracy of those trained with real EEG and non-EEG sensory data, based on the calculation 0.72/0.73. These underscore the critical role of EEG data—whether real or generated—in enhancing emotion recognition within VR environments. It also indicates the potential of generated EEG in VR settings, where collecting real EEG data may be cumbersome or intrusive.

In some cases, such as with Subjects 3, 4, 5, 8, 9, 12, and 13, classifiers using generated EEG outperformed those using real EEG. This advantage is due to CWGAN-GP's ability to generate high-quality signals with reduced noise, thereby improving classifier performance.

These results suggest that in specific instances, synthesized EEG signals can surpass the performance of real EEG data in VR contexts, where immersive experiences may significantly influence emotional responses. This demonstrates the potential of the

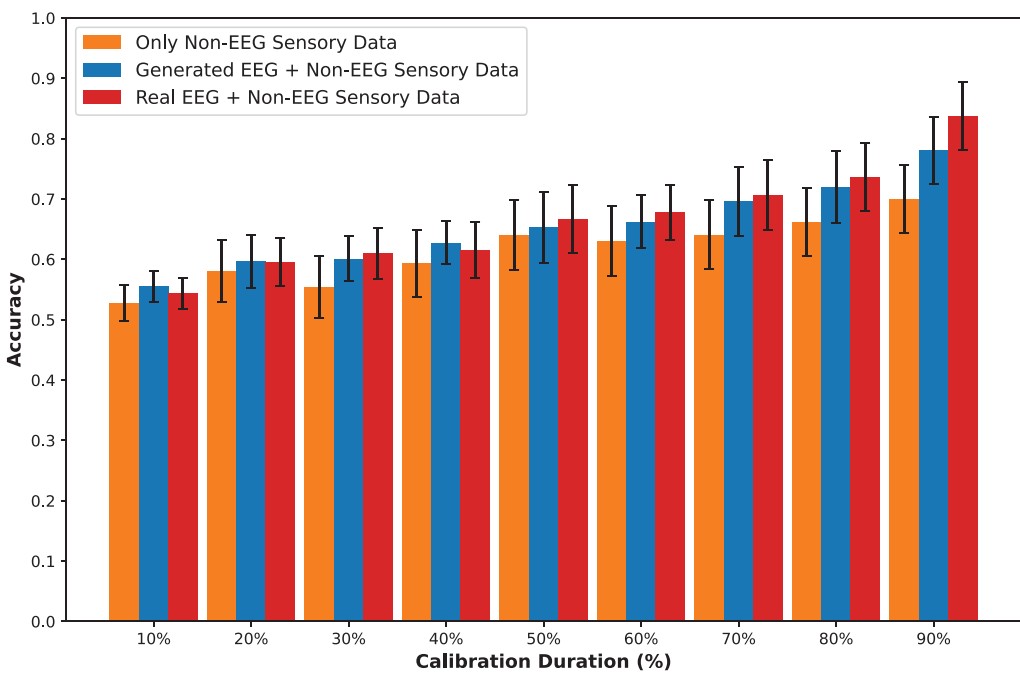

**Figure 13 Calibration duration results on GraffitiVR.**

approach to facilitate high-fidelity emotion recognition in VR settings without the need for extensive real EEG data collection.

## Results about calibration duration using GraffitiVR dataset

In Fig. 13, each bar represents the average accuracy of emotion recognition classifiers (SVM, RF, MLP), with black error bars indicating the standard deviation for individual classifiers. The y-axis denotes accuracy, while the x-axis shows the calibration duration as a percentage, relative to the actual calibration duration in the dataset. If the actual calibration length is 10 units, a calibration duration of 10% signifies that oneunit is used for both CWGAN-GP synthesized EEG generation and classifier training, with the remaining nine units used for classifier testing.

This setup examines the impact of calibration duration on emotion recognition accuracy and how it can be improved with CWGAN-GP-generated EEG signals. By varying the calibration duration—expressed as a percentage of the total calibration time—the aim was to find the optimal balance between reducing calibration time and maintaining emotion recognition accuracy.

For classifiers trained with only non-EEG sensory data (represented by the orange bar), the performance at 50% calibration (0.64) is similar to that at 80% (0.66) and 90% (0.69). This suggests that non-EEG sensory data calibration requires less time because it reaches its saturation point earlier.

For the classifiers trained with CWGAN-GP-generated EEG data and non-EEG sensory data (represented by the blue bar), the accuracy of 0.69 is achieved with 70% calibration,

indicating a CIS of 20%, compared to the accuracy of classifiers using only non-EEG sensory data at 90%.

When comparing the performance of classifiers trained with generated EEG data and those trained with real EEG, the generated EEG data achieves an accuracy of 0.65 at 50% calibration, which is acceptable compared to real EEG at 60% (0.67), 70% (0.70), and 80% (0.73). This indicates a CIS up to 30%, demonstrating the effectiveness of generated EEG data in optimizing the calibration process for emotion recognition tasks within immersive VR environments.

## DISCUSSION

Comprehensive experiments utilizing diverse datasets have shown that synthetic EEG data, derived from alternative modalities such as eye movements from the SEED-V dataset, EOG, EMG, GSR, RESP, BVP, SKT from the DEAP dataset, as well as head movements from the GraffitiVR dataset, can be as effective for emotion recognition as real EEG data. This finding underscores the potential of adversarial networks in synthesizing high-fidelity signals for complex applications.

Quantitative evaluations of the generated EEG data using metrics such as Euclidean and Wasserstein distances, as well as statistical tests like KL Divergence and T-Test analysis, confirm that the synthetic data closely aligns with real EEG signals in both temporal and frequency domains. Visualizations of "good" (low error) and "bad" (high error) examples provide additional evidence of the neuroscientific validity of the generated data, demonstrating the model's capability to replicate meaningful neural patterns essential for emotion recognition. This combination of statistical metrics and visual analysis establishes a robust validation framework for the generated data.

The core focus of our research is on utilizing synthetic EEG data to notably reduce the calibration times required in EEG-based emotion recognition systems. By employing GANs to generate EEG from non-EEG data, we effectively bridge the gap between the rich, direct neural activity measurements provided by EEG and the more accessible non-EEG modalities. This approach not only maintains the detailed feature representation necessary for high-fidelity emotion recognition but also addresses the practical limitations of traditional EEG systems, particularly the extensive calibration periods they typically require. This makes the use of EEG data more practical for real-world applications.

The results also highlight the adaptability of the proposed approach across multiple datasets with varying input modalities and spatial configurations of EEG channels. For example, the model successfully generated synthetic EEG signals for SEED-V and DEAP datasets, which include all available channels (62 and 32, respectively), and for GraffitiVR, which uses six prefrontal channels. This adaptability demonstrates the potential for leveraging non-EEG sensory data to create accurate, dataset-specific synthetic EEG signals.

The implications of this research are significant. By reducing calibration times, the door is opened for more agile and adaptable emotion recognition systems that can be deployed in real-world scenarios where time constraints and the impracticality of extensive data collection have previously been limiting factors. This is particularly valuable for user

groups such as children and the elderly, or in clinical settings where patient comfort and cooperation can be challenging.

In addition, this approach advances the use of emotion recognition in virtual reality environments. With rapid setup times, users can enjoy a seamless and immersive experience without the burden of lengthy preparation processes, enhancing the applicability of VR in education, therapy, gaming, and other fields.

Despite these advancements, the study also reveals areas for future improvement. While dataset-specific models were employed to maximize performance, the development of a single, generalizable model capable of handling diverse non-EEG inputs across datasets remains an important direction for further research. Additionally, the inclusion of richer non-EEG sensory data could enable more comprehensive cross-modality testing and enhance the generalizability of the generated signals. These limitations provide a roadmap for future work to refine the framework and expand its applicability to more complex real-world scenarios.

## CONCLUSIONS

This study has presented a novel approach to streamline the calibration process in BCI-based emotion recognition systems by leveraging heterogeneous adversarial transfer learning. We have demonstrated that synthetic EEG signals generated from non-EEG modalities can closely emulate the statistical and morphological properties of real EEG data, effectively bypassing the traditionally time-intensive calibration phase. This approach bridges the gap between the rich neural activity measurements provided by EEG and the more accessible non-EEG data, offering a practical solution for real-world applications.

The method's validation across diverse datasets, including SEED-V, DEAP, and GraffitiVR, highlights its robustness and versatility in handling various input modalities and spatial configurations. Quantitative metrics such as Euclidean and Wasserstein distances, alongside visualizations of low- and high-error examples, provide strong evidence of the model's ability to replicate essential neural patterns relevant to emotion recognition. These results not only establish the validity of the generated data but also underscore its neuroscientific relevance for emotion-related studies.

Furthermore, the findings reveal that the synthetic EEG signals maintain a balance between fidelity and variability, ensuring meaningful feature representation while avoiding overfitting. This characteristic enhances the generalizability of the generated data, paving the way for emotion recognition systems that are both accurate and adaptable. By reducing calibration times, the proposed framework has the potential to make EEG-based systems more practical for real-world scenarios, particularly in time-sensitive or user-constrained environments such as healthcare, education, and virtual reality applications.

Despite these advancements, certain limitations remain. The study employs dataset-specific models to optimize performance, and further work is needed to develop a single, generalizable model capable of handling diverse non-EEG inputs across datasets. Additionally, the inclusion of richer non-EEG sensory data could enhance the

cross-modality capabilities of the framework. Future research will also focus on refining the model to capture more complex neural dynamics and exploring its applicability to real-time systems and multimodal emotion recognition frameworks.

In summary, this research lays a strong foundation for the use of synthetic EEG signals in emotion recognition and other BCI applications. By demonstrating that high-quality synthetic data can replace traditional EEG recordings in key tasks, this work marks a significant step forward in making EEG-based systems more accessible, efficient, and adaptable to various practical contexts.

## LIMITATIONS AND FUTURE WORK

While the proposed HATL framework shows promising results, several limitations should be considered. Firstly, the datasets used in this study—SEED-V, DEAP, and GraffitiVR—each contain unique combinations of EEG channels and non-EEG sensors, which can reduce the generalizability of the HATL architectures. The limited availability of non-EEG sensor data in these datasets further restricts the extent to which multimodal inputs can be leveraged within the framework.

Moreover, the lack of extensive open-source multimodal datasets that are compatible with this framework constrains the scope of the research. In VR environments specifically, it is challenging to deploy multiple sensors, which limits the practical feasibility of generating EEG data from diverse non-EEG sources.

Additionally, each dataset uses different emotional labels, necessitating that we treat each dataset individually in this study. This variability in emotional labels hinders a unified approach to training and evaluating the model across datasets, posing a limitation to the scalability and generalizability of the framework.

In future work, we plan to develop zero-shot calibration frameworks designed for VR environments. This initiative will focus on enhancing generalization capabilities across different subjects and adapting to various environments, both in 2D and VR. This will involve exploring methods to improve the flexibility of emotion recognition systems, enabling them to work seamlessly with new users and unfamiliar settings. By emphasizing these aspects, we aim to create more robust and adaptable frameworks for BCI-based emotion recognition.

## LIST OF ABBREVIATIONS

| | |
|---|---|
| **BCI** | Brain-computer interface |
| **CIS** | Calibration improvement score |
| **CGAN** | Conditional generative adversarial network |
| **CWGAN** | Conditional Wasserstein generative adversarial network |
| **CWGAN-GP** | Conditional Wasserstein generative adversarial network with gradient penalty |
| **EEG** | Electroencephalography |
| **GAN** | Generative adversarial network |
| **HATL** | Heterogeneous adversarial transfer learning |
| **VR** | Virtual reality |

### Funding
The authors received no funding for this work.

### Competing Interests
The authors declare that they have no competing interests.

### Author Contributions
- Mehmet Ali Sarikaya conceived and designed the experiments, performed the experiments, analyzed the data, performed the computation work, prepared figures and/or tables, authored or reviewed drafts of the article, and approved the final draft.
- Gökhan Ince conceived and designed the experiments, authored or reviewed drafts of the article, and approved the final draft.

### Data Availability
The code scripts and raw data are available in the Supplemental Files.

The third party datasets are available at:

- SEED-V dataset: https://bcmi.sjtu.edu.cn/~seed/seed-v.html
- DEAP dataset: https://www.eecs.qmul.ac.uk/mmv/datasets/deap.
- GraffitiVR dataset https://www.emerald.com/insight/content/doi/10.1108/arch-03-2023-0087.

### Supplemental Information
Supplemental information for this article can be found online at http://dx.doi.org/10.7717/peerj-cs.2649#supplemental-information.

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
