# Peer review of "Improved BCI calibration in multimodal emotion recognition using heterogeneous adversarial transfer learning"

_PeerJ Computer Science, doi:10.7717/peerj-cs.2649_

## Round 0.1 · original submission · Major Revisions

Dear authors,

Thank you for submitting your article. Feedback from the reviewers is now available. Your article has not been recommended for publication in its current form. However, we do encourage you to address the concerns and criticisms of the reviewers and resubmit your article once you have updated it accordingly. Before submitting the revised paper, following should also be addressed:

1. Please pay special attention to the usage of abbreviations. Spell out the full term at its first mention, indicate its abbreviation in parenthesis and use the abbreviation from then on.
2. Equations should be used with correct equation number. Many of the equations are part of the related sentences. Attention is needed for correct sentence formation.
3. All of the values for the parameters of all algorithms should be given.
4. Advantages, disadvantages and limitations of the proposed method should be provided.

Best wishes,

·

Basic reporting

This paper has high-quality scientific content with significant original contributions in professional and concise English. The article is well-written and logically structured, linking the causes and effects and providing generous references. The paper reveals the implementation of GAN architecture (artificial intelligence) for generating synthetic EEG data aimed at improving the calibration time necessary for emotion recognition using a brain-computer interface. The paper includes technical details related to mathematical formulas underlying deep learning.

Experimental design

There were presented the results obtained by comparing three experimental paradigms. There were included graphical representations, tables that were detailed discussed.

Validity of the findings

I appreciate the content of this paper, as well as the novel ideas and the rich background information. I am still interested into finding out more about how the synthetic EEG data was generated. Could you please explain more insights about the methods you applied to generate the EEG data starting from the non-sensory data? This is a new approach and I consider that many researchers would be interested in it. Also, which is the EEG headset you used for the VR Graffity application? How could you integrate the EEG data acquisition and monitoring with the VR technology? What are the customized or official software platforms that you used? Moreover, please clearly specify if the VR Graffity application is your original contribution. If so, then probably you can add a picture with its graphical user interface that can improve the design of your paper and it may capture the interest of novice researchers who gathered programming skills in both developing brain-computer interfaces and VR software applications.
Congratulations on your achievement and I wish you good luck with your future research projects.

Reviewer 2 ·

Basic reporting

The authors propose a strategy to reduce the calibration time and data requirements for BCIs based on multimodal emotion recognition. The main idea is to use a variation of conditional Wasserstein generative adversarial network to produce EEG data with emotional content with non-EEG data.

The introduction and background provide a proper contextualisation of the problem. I suggest only adding a definition of “calibration” for non-experts in the field of brain-computer interfacing, and providing not only the strategies proposed by the literature in terms of methodology, but also the obtained results in a quantitative manner and specifying the classification tasks.

The paper is generally well-written, but there are some phrases that need to be revised, such as in line 21, 68-71, 512 (I think the Authors refer to a “leave-one-subject-out” strategy but refer to it with “one-leave-out”).

The structure conforms to PeerJ standards. The figures are relevant, high quality, well-labelled and described.

The reference to the publicly available data is clearly provided and the codes are well documented.

Experimental design

The research is in the scope of the journal and the guiding research question clear. The people involved in BCI research are always striving to have more data to use for system calibration or to have a better understanding of the data. I am very thankful to the Authors for giving more visibility to such an important topic.

While the idea is very interesting and provides a good starting point to generate emotion-related data, it is not clear the validity of these new data generation.
I understand that the main feature of GAN-based methodologies is to completely rely on the feedback mechanism between the generator and discriminator to refine the synthetic data, however it would be important to provide a clear assessment of the differences and similarities between the real and synthetic data, even through the visualization of the time series.
In this case, a figure presenting some “good” and “bad” examples of generated data is missing.
Considering that one of the main features of the proposed methodology is the ability of generating EEG data from non-EEG ones, it would be best to provide details on how it is done and if these synthetic data could be really taken for real data by an expert in the field.
It could be possible that the model generates data that are able to provide emotional content but that are extremely redundant or introducing excessive bias.

How do the Authors ensure that the produced data are neuroscientifically compliant? Do the resulting signals present similar patterns to real EEG data? Is the process performed in a completely black-box manner?

Considering the application of the methodology, I have some concerns regarding data management.

It is not clear in the data description section if the data are pre-processed and which portion of the signals are considered.
Regarding the EEG signals, are all the available channels used? If not, which are the selection criteria? Are one or more channels generated by the proposed model?
Are the non-EEG data used as time series or as features only? Is there a clear difference between the use of different non-EEG data for EEG signal generation? Does the multi-modality claim hold for all the performed experiments?

For what I understand, a dedicated model is instantiated for each of the used datasets. Could a generalizability of the model be declared?
How are the different emotional labels handled?

Validity of the findings

The results are well described, but the doubt regarding the initial claim on the production of EEG data from non-EEG ones remains, considering that no analysis of the generated data is provide in terms of signal morphology and time and/or frequency characteristics. Is an entire EEG recording produced considering all the possible available channels? Which are the differences between the original and generated data? Do they maintain similar patterns to real EEG signals, while introducing a certain degree of variability? How does the generative method produce cleaner signals starting from noisy ones?

Another concern regards the statement (line 547) related to the presence of balanced data in terms of emotional content. Usually, a data exploratory analysis is performed before using them as inputs to learning models to understand the class distribution. Why is the statement regarding class balance provided as a consequence of the obtained results?

·

Basic reporting

1. Authors must show explain the novel contribution of the work with proper justification of the outcomes. Novel Contribution of the proposed work can be added at the end of Introduction.
2. The abstract can be improved and the outcome of the work in terms of achieved performance calculations must be included in the abstract.
3. Literature survey is missing and need to be modified based on current state of art methods. Some more paper based on current study in deep learning model multimodal emotion recognition.

Experimental design

4. The computational complexity of the algorithm must be discussed. Also, compare the proposed method in terms of computational complexity?
5. Future work and limitations of the proposed work can be added and discussed.
6. Comparative analysis with respect to various performance metrics is missing? The comparison can be a bit unfair if different data is not used for comparative analysis.
7. Has the Author implemented the architecture from scratch and identified the novel condition in deep networks.
8. Specification of the implementation platform is missing.
9. Precision vs. recall curves of the proposed algorithms with respect to data sets must be included.
10. Comparative analysis of various performance parameters with respect to data sets and ground truth data sets must be discussed.

Validity of the findings

11. Limitations of the proposed work can be added and discussed.
12. In all results tables’ utilized datasets like in table 2 must be cited with proper and specific citations.
13. Various abbreviations also must be included.
14. How much data should be considered for training and testing for architecture implementation? Details of training and testing data sets must be tabulated.
15. Comparative analysis with respect to real-time time analysis is missing?
16. Layers details of proposed architecture must be included.
17. Loss vs. epochs and Accuracy vs. epoch’s graphs for various tested model must be added.

---

## Round 0.2 · Minor Revisions

Dear Authors,

According to Reviewer 1 the manuscript still needs improvement. Please address the comments of Reviewer 1 and resubmit your paper.

Best wishes,

Reviewer 2 ·

Basic reporting

I thank the Authors for providing precise responses to my concerns.

The manuscript has greatly improved, and some methodological aspects have been clarified.

I have some additional suggestions to make before suggesting a final acceptance of the paper, but overall, they would mainly involve only adding a slightly modified version of your response to the manuscript.

Summary:
- Clear and unambiguous, professional English is used throughout.
- The literature references and the article structure are adequate.
- There are some minor changes to make on some figures.
- Some changes should be applied to ensure a correct undestanding of the provided results.
- All the required definitions are present.

Experimental design

In the following are reported the final two major concerns on the experimental design, which aligns with the aims and scope of the journal.

In response 4 you have explained that the features of non-EEG data are used for synthetic EEG data generation. I would highly recommend highlighting it directly in the “System Architecture” section to avoid any misunderstanding.
In fact, now it seems that the generator produced EEG signals, thus time series, more than patterns, thus feature-based information.
This further justify the presence of frequency-based brain activation images rather than the time series/signals to make comparisons.
I think you can directly leverage on your response: “Our model generates synthetic data across all specified channels for each dataset based on the non-EEG sensory input. Non-EEG data are treated as extracted features rather than continuous time series. Each non-EEG modality undergoes feature extraction based on statistical and frequency-domain attributes. For instance, eye-tracking features in SEED-V include pupil dilation, saccades, and fixation duration, while GraffitiVR utilizes head movement features. This feature-based approach optimizes the model’s capacity to capture key aspects of each modality relevant to EEG generation”.

Concerning the classification task, I think you should make a report on the labels as per the response, for example in the “Experimental Framework”, specifying that all the four quadrants of the valence/arousal plane are considered for two datasets, while some specific categorical emotions are present for one of the three. For what I understand from the previous sections, the outcome is binary for the different conditions. Is that correct? Please, clarify it to ensure that the readers would understand that the results are an average performance between different binary tasks.

Validity of the findings

Concerning the validity of the findings, there are just a couple of changes to apply.

For Figure 2, I suggest having the same range for the distance on the y-axis for each distance metric. For example, the plots of the Euclidean distance could have a y-axis with a value range starting from 1.0 and stopping to 13.5.

For Figure 3 to 8, the colour code should be explained, and the colour-bar appear to ensure that the representations are all in the same range of values. Moreover, I suggest adding to the captions the name of the dataset the images refer to.
Do the frequency representations refer to specific emotions or time frames?

In the discussion/conclusion I would also suggest adding some insightful assessment of your work and result, that you have already reported in responses 3, 4, and 5. This would help wrapping up your report.

Additional comments

I have no further comments to make.

·

Basic reporting

all my comments and concerns has been added successfully. i accept in current form,

Experimental design

N/A

Validity of the findings

N/A

Additional comments

all my comments and concerns has been added successfully. i accept in current form,

---

## Round 0.3 · accepted · Accept

Dear Authors,

Thank you for clearly addressing the reviewers' comments. Your paper seems sufficiently improved and ready for publication.

Warm regards,

Reviewer 2 ·

Basic reporting

I thank the Authors for addressing all my concerns.

I suggest an acceptance of the paper for publication, considering that:
- A proper explanation of the data generation has been provided.
- The classification tasks have been highlighted and clarified.
- The figures supporting the claims related to the results have been clearly described and modified.
- The take on messages regarding the validity of the findings and the overall assessment of the performed experiments reported.

I have no further comments, besides wishing the Authors all the best and thank them for the good work.

Experimental design

no comment

Validity of the findings

no comment

Additional comments

no comment